# Integrated photonic encoder for low power and high-speed image processing

Xiao Wang[1,6], Brandon Redding [2,6], Nicholas Karl [3], Christopher Long[3], Zheyuan Zhu [4], James Skowronek [1], Shuo Pang[4], David Brady[1] ✉ & Raktim Sarma [3,5] ✉

Modern lens designs are capable of resolving greater than 10 gigapixels, while advances in camera frame-rate and hyperspectral imaging have made data acquisition rates of Terapixel/second a real possibility. The main bottlenecks preventing such high data-rate systems are power consumption and data storage. In this work, we show that analog photonic encoders could address this challenge, enabling high-speed image compression using orders-of-magnitude lower power than digital electronics. Our approach relies on a silicon-photonics front-end to compress raw image data, foregoing energy-intensive image conditioning and reducing data storage requirements. The compression scheme uses a passive disordered photonic structure to perform kernel-type random projections of the raw image data with minimal power consumption and low latency. A back-end neural network can then reconstruct the original images with structural similarity exceeding 90%. This scheme has the potential to process data streams exceeding Terapixel/second using less than 100 fJ/pixel, providing a path to ultra-high-resolution data and image acquisition systems.

From the invention of photography until the early 1990, the function of a camera was to record analog images. With the development of solid-state focal planes and digital coding, however, this function changed, and modern digital cameras act as transceivers that transform massively parallel optical data streams into serial coded electronic data that is processed 1 pixel at a time[1-3]. While this paradigm shift introduced numerous advantages, the power consumption associated with electronic digital processing, along with limits on data transmission rate and storage capacity, are now the major bottlenecks limiting image data acquisition rates[1-4]. In contrast, compact lens designs are capable of resolving greater than 10 gigapixels of transverse resolution[5,6], while advances in multimodal imaging systems capable of acquiring spectral, polarization, temporal, and range information could enable future imaging systems to acquire (Tera)$10^{12}$ pixels per second of data. However, the power consumption and resulting coding capacity of opto-electronic transceivers is one of the primary barriers to achieving such systems[1-3].

In current digital electronics-based image processing systems, electrical power consumption is proportional to the number of mathematical operations performed on each pixel[1]. Conventional image signal processing (ISP) systems perform 100-1000 operations per pixel to first condition (e.g., compensate for pixel non-uniformity, hot-pixels, denoising etc.) and then compress the image data stream, resulting in a per pixel cost of 0.1-1 microjoule[1]. Refs. 1–3. show that in high-resolution gigapixel imaging systems operating at 30 frames/s, the image sensors draw 100 milliwatts/megapixel, whereas the back-end digital image processing pipeline draws approximately 10X more power amounting to ~1000 milliwatts/megapixel. This implies that processing 1 terapixel of data would result in power consumption of around 1 megawatt, which is prohibitive for many applications. In

[1]Wyant College of Optical Sciences, University of Arizona, Tucson, Arizona, USA. [2]U.S. Naval Research Laboratory, Washington, DC, USA. [3]Sandia National Laboratories, Albuquerque, New Mexico, USA. [4]CREOL, The College of Optics and Photonics, University of Central Floria, Orlando, Florida, USA. [5]Center for Integrated Nanotechnologies, Sandia National Laboratories, Albuquerque, New Mexico, USA. [6]These authors contributed equally: Xiao Wang, Brandon Redding. ✉e-mail: djbrady@arizona.edu; rsarma@sandia.gov

addition, while image compression is required for most remote sensing applications, many of these pixel conditioning operations are performed at the front end regardless of whether they are necessary for a given application. Recently, different ISP approaches, such as blind sensor head compression were proposed to reduce the number of operations per pixel[1]. Blind compression, in this context, can be understood as implementing the first layer or two of a deep neural network-based auto-encoder on the read-out data stream. While this approach substantially reduces the number of operations per pixel, the per pixel power cost still remains unacceptably high, at ~0.01-0.1 microjoule[1].

Future Terapixel/second imaging systems (e.g., imaging systems with (Giga)$10^9$ -pixel resolution with frame rates of (Kilo) $10^3$ Hz or imaging systems with higher pixel resolution but slower frame rates) will require an alternative ISP approach with dramatically higher throughput and lower power consumption. In this work, we explore the potential for analog photonics to improve throughput and power consumption in an image acquisition pipeline after image formation on the focal plane array. This is in stark contrast to the significant body of work focused on image classification, inference, or compressed sensing of the raw scene information that operates by processing the scene data directly in the analog optical domain at the image acquisition stage[4,7–11]. While both schemes are valuable, we believe there are several important advantages for our approach, which has been relatively under-explored. First, conventional imaging optics and focal plane arrays are highly optimized, and it is difficult to improve on their baseline performance, particularly in wavelength regimes where high-resolution focal plane arrays are available. As a result, our approach does not attempt to alter the original image formation process. Instead, we designed an accelerator to address the two main bottlenecks limiting persistent, high-data-rate image acquisition: power consumption and compression speed. Second, by positioning the accelerator after the image formation and initial optical-to-electrical conversion step, this scheme is immediately compatible with any image acquisition system, regardless of operating wavelength, camera resolution, front-end optics, frame rate, or application. There is no added insertion loss before the initial detection stage, enabling compatibility with low-light and high-speed imaging applications. Perhaps most significantly, our approach works with either ambient illumination imaging or active illumination, whereas many of the compressive imaging schemes that operate at the image acquisition stage require active illumination with a coherent source[7,8], which severely limits the application space.

Instead of modifying the original image formation process, our approach builds on the neural network framework proposed in the blind compression work[1], which allows the key front-end image processing task to be accomplished using a single matrix-vector multiplication. Fortunately, analog optical computing is particularly well-suited for this type of operation and is being explored for a variety of matrix-multiplication-heavy computing applications[12–15]. The key advantage is that optical computing engines can perform matrix multiplication with energy consumption that scales linearly with the dimension of the input dataset ($N$) as opposed to the quadratic scaling ($N^2$) inherent to electronic approaches. In addition, optical computing engines are able to process $N$ pixels in parallel with an overall speed that can exceed 100 GHz, only limited by the speed of the optical modulators and photodetectors used to encode the input data and record the result[16–18].

In this work, we show that an optical image processing engine can take advantage of these unique features to enable high-speed image compression with the potential for orders-of-magnitude lower power consumption than current techniques. Our approach is based on a passive, CMOS-compatible silicon photonics device that performs the matrix-vector multiplication required for front-end image processing. We experimentally demonstrate image compression with a ratio of 1:4

and develop a back-end neural network capable of reconstructing the original images with an average peak signal-to-noise ratio (PSNR) ~ 25 dB and structural similarity index measure (SSIM) ~ 0.9, comparable to the common electronic software-based lossy compression schemes such as JPEG[19]. By processing modest-sized kernels in series, our approach is inherently scalable and could process high-frame rate, limited region-of-interest image data, or Gigapixel images at slower data rates. Finally, by constructing the optical image compression engine on a silicon photonics platform, this approach meets the size-weight-and-power and integration requirements for a wide range of applications including surveillance, microscopy, machine vision, astronomy, or remote sensing. Analysis of the throughput and power consumption for our optical image processing engine indicates that this technique has the potential to encode Terapixel/second data streams utilizing <100 femtojoules per pixel−representing a > 1000x reduction in power consumption compared with the state-of-the-art electronic approaches. This improvement in throughput and power consumption is made possible by parallelization of pixel processing and transferring the majority of the image processing and conditioning tasks (e.g., denoising, linearization, etc.) from the front-end encoding interface to the back-end decoding interface. This trade-off is particularly attractive for remote sensing and imaging applications in which image reconstruction is only performed on a need-to-know basis and is usually conducted at a remote cloud site with better access to power.

## Results
### Operating principle of the image encoder
The optical image compression scheme relies on an auto-encoder neural network framework in which the compressed image is naturally formed at the "bottleneck" layer in a neural network[20]. This approach has gained traction in recent years due to its ability to simultaneously perform data compression, dimensionality reduction, and denoising[20–22]. In our implementation, the first half of the neural network (mapping the original image data to the compressed image at the bottleneck layer) is performed optically, while the second half of the network (reconstructing the image) is performed using digital electronics. A schematic of the optical encoder and corresponding neural network structure is shown in Fig. 1. Our hybrid optoelectronic auto-encoder takes advantage of the fact that most neural encoding networks are not very sensitive to the details of the feature map (i.e., the weights and connections) implemented by the first few layers. In fact, researchers have shown that the first few layers can often be assigned random weights without compromising performance[23]. This allows us to use a pre-designed, passive photonic device to perform the transform used in the first layer of the auto-encoder network. In this case, we designed the photonic layer to perform local kernel-like random transforms on small blocks of the image at a time. This random encoding scheme was selected based on compressive measurement theory, which has shown that random transforms are ideal for a variety of dimensionality reduction and compression tasks[24–28]. However, unlike conventional compressed sensing measurements, which are applied at the data acquisition stage[11], we are addressing the problem of dimensionality reduction and data compression after image formation.

As shown in Fig. 1, the silicon photonics-based image encoder consists of $N$ single mode input waveguides, each with a dedicated modulator, followed by a multimode waveguide region, a random encoding layer, and $M$ photodetectors. A laser (not shown) provides light with equal amplitude to the $N$ input waveguides where the $N$ modulators encode a $\sqrt{N} \times \sqrt{N}$ pixel block of the input image onto the amplitude of light transmitted through each waveguide. Light from each waveguide is then coupled into a multimode waveguide region before scattering through the random encoding layer and finally reaching the photodetectors. The random encoding layer consists of a

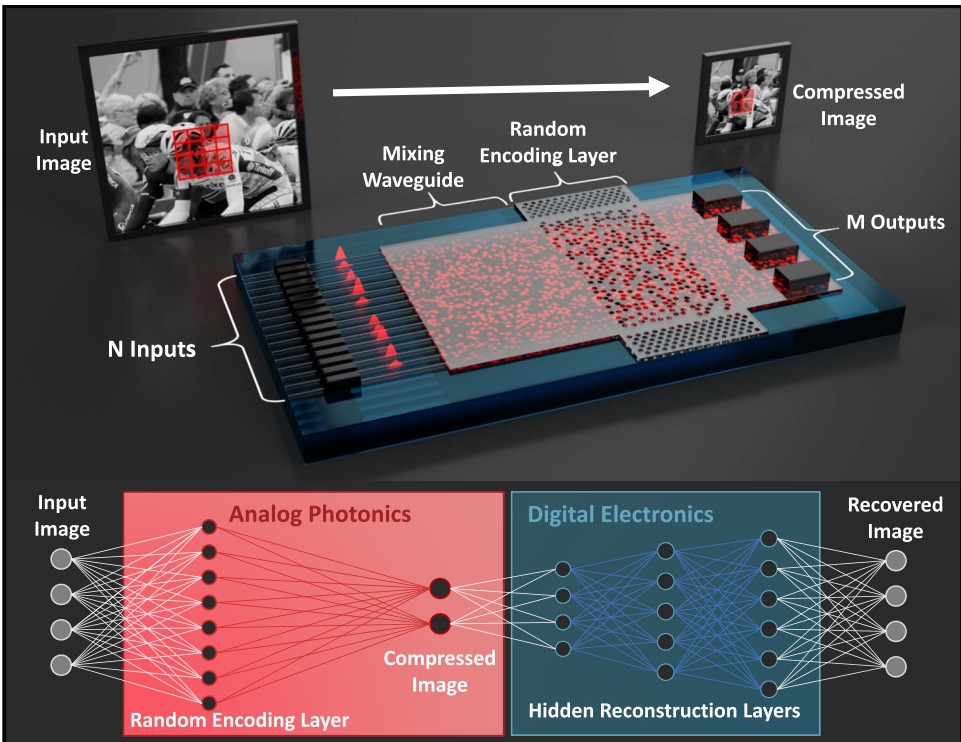

**Fig. 1 | Working principle of the photonic image encoder.** The silicon photonics-based all-optical image encoder consists of a series of $N$ single mode input waveguides which carry information of the pixel of the images in the optical domain representing a $\sqrt{N} \times \sqrt{N}$ pixel block of the input image. The input waveguides connect to a multimode silicon waveguide region followed by a disordered scattering region which encodes the input through a local random transformation for image compression. The encoded output, which is a random looking spatially varying intensity pattern, is binned into $M$ non-overlapping spatial regions (corresponding to $M$ detectors) such that $M < N$ and the entire image is compressed through a series of these block-wise transforms. While the compression is performed optically, the reconstruction and image conditioning steps are performed electronically at the backend.

series of randomly positioned scattering centers fabricated by etching air holes in the silicon waveguiding layer (see the "Methods" section for more details). Since the optical device operates in the linear regime, we can describe the encoding process using a single transmission matrix ($T$), which relates the input ($I$) to the transmitted output ($O$) as $O = TI$, where $I$ is a $N \times 1$ vector, $O$ is a $M \times 1$ vector, and $T$ is a $M \times N$ matrix. By forcing $M$ to be less than $N$, the device effectively performs a single matrix multiplication in order to compress an $N$ pixel block of the original image into $M$ output pixels. Since the random encoding layer is entirely passive, this compression process can be extremely fast, operating on $N$ pixels in parallel at speeds limited only by the modulators and photodetectors. In addition, the energy consumption scales linearly with $N$ (i.e., the number of modulators), even though the device performs $M \times N$ operations.

To further clarify the role of our data compression system, we can consider the entire image acquisition/compression process in 4 steps:

(1) Conventional imaging optics form an image on the focal plane array of the camera.

(2) Conventional focal plane array detectors convert the analog optical image to the electrical domain.

At this point we have two options:

(3a) Most commercially available cameras are designed to digitize the image data recorded on the focal plane array. Using this type of camera, we would then use a digital to analog converter (DAC) to drive the optical modulators on-chip, re-encoding the image information in the optical domain on an optical carrier.

(3b) Focal plane arrays are also commercially available, which provide a direct analog output[29]. This analog output could be used to directly drive the optical modulators to re-encode the image

information without an intermediate digitization step. A transimpedance amplifier (TIA) can directly convert the analog photocurrent to voltage to drive the re-encoding modulation, with appropriate amplification.

(4) The on-chip photonic encoder then performs high-speed, low power compression and the output from the detectors on-chip is digitized and stored for off-line image reconstruction.

Using focal plane arrays that provide an analog output (option 3b) has the potential to significantly reduce the overall power consumption and throughput by avoiding the intermediate analog to digital conversion (ADC)/DAC steps. However, our underlying approach (the photonic chip performing compression) is compatible with both approaches, which is useful given the ubiquity of cameras with integrated ADCs.

The local kernel size, $N$, is a key parameter driving the performance of the photonic image processing engine. While using smaller kernel-like transforms reduces the data throughput, since the device can now only compress $N$ pixels at a time, smaller kernels also have several advantages. First, local transforms maintain the spatial structure of the original image, which tends to improve the image reconstruction, as we discuss in the next section. Second, the kernel approach can be used to compress arbitrarily large images without requiring a corresponding increase in the numbers of modulators and detectors. Third, using these local transforms helps to isolate noise from a given pixel (e.g., a hot pixel), which could otherwise spread across the entire compressed image. Finally, since this compression scheme effectively maps the input image blocks to speckle patterns, using a large kernel could lead to low-contrast speckles which can also degrade the image reconstruction similar to the trend observed in speckle-based spectrometers[30].

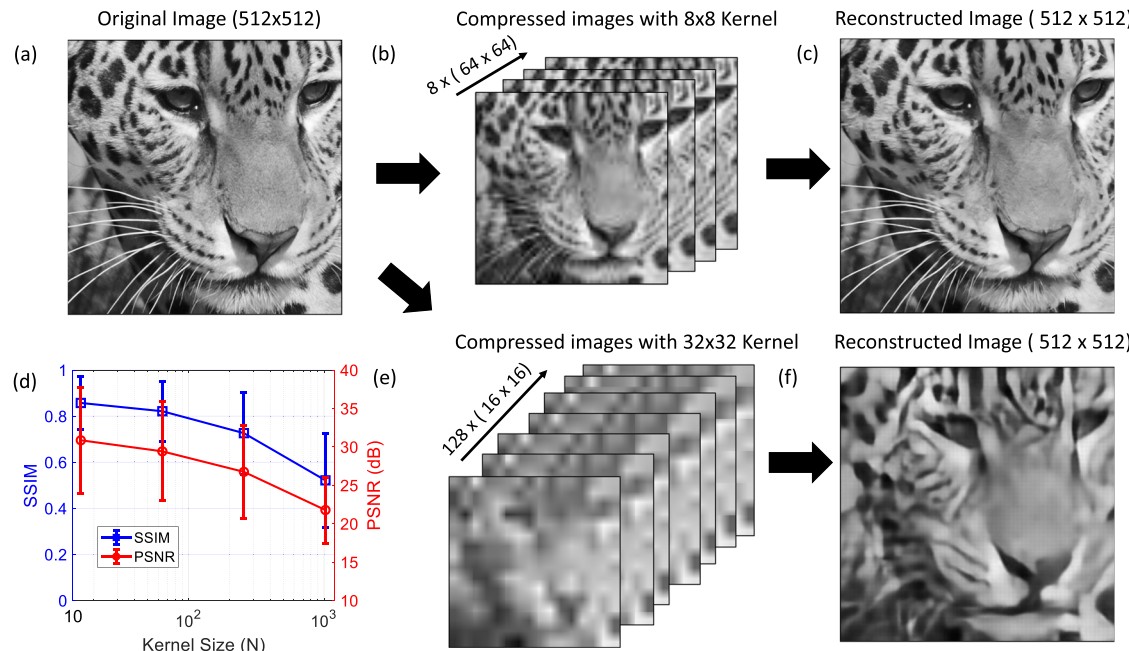

**Fig. 2 | Effect of kernel size on image compression and reconstruction using images from the DIV2K and Flickr2K dataset[31,32]. a** One of the original (512 × 512 pixel) grayscale noiseless images taken from the DIV2K and Flickr2K dataset that is used to demonstrate compression. **b, e** Compressed images using an 8 × 8 (**b**) and 32 × 32-pixel kernel (**e**). For both cases, the compression ratio (M:N) is fixed to be 1:8 where in (**b**) $N = 64$, and $M = 8$ and in (**e**) $N = 1024$, and $M = 128$. The sizes of the compressed images are shown in brackets (**c, f**) Reconstructed images (512 × 512-pixel) from the compressed images shown in (**b**) and (**e**) respectively. **d** Mean PSNR (red) and SSIM (blue) of the reconstructed images from the test data as a function of the kernel size. The error bars correspond to 1 standard deviation.

## Effect of kernel size and kernel type on image compressibility

To determine the effect of kernel size on image compression, we performed numerical simulations of the image compression and reconstruction process using images taken from the DIV2K and Flickr2K dataset[31,32] and synthetically generated random T matrices. To reduce the computation time, the images were converted to grayscale and cropped to a resolution of 512 × 512 pixels. The dataset consisted of 4152 images divided into a 3650-image training dataset and a 502-image validation dataset. In this case, the compression process was simulated by multiplying each $\sqrt{N} \times \sqrt{N}$ block of an image by a numerically generated random matrix consisting of real, positive numbers uniformly distributed between 0 and 1. We then trained a neural network to reconstruct the original image from the compressed image (see "Methods" for a detailed description of the neural network architecture and training routine). Finally, we used the test images from DIV2K and Flickr2K dataset to evaluate the reconstructed image fidelity after compression using kernels of varying sizes.

An example image from the set of test images is shown in Fig. 2a along with the compressed images obtained using 8 × 8 (Fig. 2b) and 32 × 32-pixel kernels (Fig. 2e). In this case, we fixed the compression ratio (M : N) at 1:8 and the images were compressed from 512 × 512 to 8 × (64 × 64) or 128 × (16 × 16) pixel datacubes. The reconstructed images using the two kernel sizes are shown in Fig. 2c, f. Using a smaller kernel size clearly retains more of the spatial structure in the compressed image (Fig. 2b), resulting in a higher fidelity reconstruction. The average peak signal-to-noise ratio (PSNR) and the structural similarity index measure (SSIM) of the reconstructed images in the test image dataset are shown in Fig. 2d as a function of kernel size. We found that smaller kernels generally result in higher-quality image reconstruction. Unlike spatially uncorrelated and sparse data—which can be efficiently compressed using large random matrices—image data naturally includes spatial structure and spatial correlations which are important to maintain. In general, the optimal kernel size will depend on the type of images being compressed and will be impacted

by factors such as the sparsity and spatial frequency content of the images.

For benchmarking, we compared our compression technique using an 8 × 8 kernel and 16 × 16 kernel to standard JPEG compression (which uses an 8 × 8 kernel) on images from the same test dataset (DIV2k and Flickr2K). Figure 3 (a, b) compares the average PSNR (a) and SSIM (b) of the test image dataset obtained using JPEG compression to our encoding scheme as a function of compression ratios. As shown in Fig. 3a, b, our approach provides slightly lower PSNR/SSIM than JPEG at low compression ratios (e.g., <1:16) and comparable PSNR/SSIM at intermediate ratios (1:32 and 1:64). Our approach also enables higher compression ratios than is possible using JPEG (which is limited to a ratio of 1:64) and maintains an acceptable average PSNR > 20 dB up to ratios of 1:256. Furthermore, unlike photonic compression which allowed for fixed compression ratios, not all images from the test dataset could be compressed to the ratio of 1:64 using JPEG compression (only ~400 out of 500 test images could be compressed up to 1:64). Figure 3c, d shows the compression ratio dependence of the same example image as in Fig. 2a reconstructed after compression using the photonic approach and using JPEG. For this image, the highest compression ratio that we could achieve using the JPEG algorithm is 1:45 (see Supplementary Information: Section S6. Comparison of JPEG and photonic compression for more examples). At ratios of 1:8 and 1:16, both techniques provide excellent quality images. At higher compression ratios (compression ratios > 1:32), JPEG compression introduces pixilation artifacts, whereas the photonic compression scheme loses some of the higher spatial frequency content. These differences result from fundamental distinctions between the two compression schemes.

The JPEG compression algorithm applies a discrete cosine transform (DCT) to every 8 × 8-pixel block in an image[19]. It then applies a thresholding operation to store the most significant basis functions. While this nonlinear, data-dependent transformation approach results in high-quality compression for low compression ratios or for sparse images, it has several drawbacks compared to the photonic

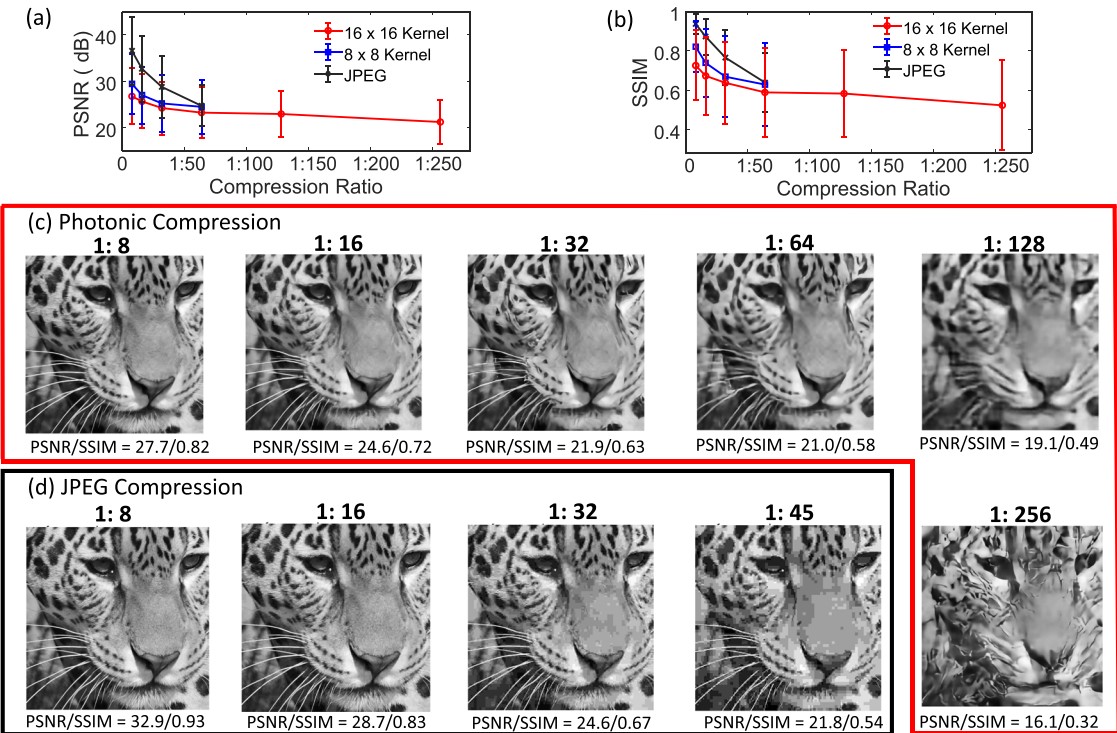

**Fig. 3 | Comparison of photonic image compression to digital JPEG image compression using images from the DIV2K and Flickr2K dataset[31,32].**
**a** Comparison of mean PSNR of the reconstructed images from the test dataset (sample size = 500 test images) and images compressed using the JPEG algorithm as a function of compression ratio. The error bars correspond to 1 standard deviation. The blue (red) line corresponds to photonic compression with an 8 × 8 (16 × 16) kernel. The black line corresponds to JPEG compression. The data for compression ratio of 1: 64 for JPEG compression has a smaller sample size of only 400 images as ~100 test images could not be compressed beyond 1:45 using the JPEG algorithm. **b** Comparison of mean SSIM of the reconstructed images from the test dataset (sample size = 500 test images) and images compressed using the JPEG algorithm as a function of compression ratio. The error bars correspond to 1 standard deviation.

The blue (red) line corresponds to photonic compression with an 8 × 8 (16 × 16) kernel. The black line corresponds to JPEG compression. The data for compression ratio of 1: 64 for JPEG compression has a smaller sample size of only 400 images as ~100 test images could not be compressed beyond 1:45 using the JPEG algorithm. **c** Reconstructed images that were compressed using the photonic compression approach using a 16 × 16 kernel with varying compression ratios ranging from 1:8 to 1:256. The original test image is shown in Fig. 2(a). The corresponding PSNR/SSIM are stated along with the images. **d** Compressed images using the JPEG algorithm with varying compression ratios ranging from 1:8 to 1:45. The original test image is shown in Fig. 2(a). The corresponding PSNR/SSIM are stated along with images. The maximum compression ratio that can be achieved using JPEG compression for this image is ~ 1:45.

compression scheme introduced here, particularly for high data-rate imaging applications. First, JPEG compression ratios are image-dependent. This can be problematic for high-data-rate image acquisition, which would have to allocate variable-sized memory blocks as the compression ratio changes. Second, JPEG performs many more operations on each pixel than our photonic scheme, since it performs a full DCT transformations on every 8 × 8 pixel block before selecting the basis functions to retain. This results in increased power consumption and slower throughput since it requires multiple clock-cycles. Finally, JPEG is unable to perform de-noising or image conditioning. As described in the Experimental section below, the photonic scheme is able to simultaneously perform image compression, image conditioning (pixel linearity, hot pixels, etc.), and denoising by using a neural-network based non-linear decoding scheme.

In addition to kernel size, we also investigated two types of random kernels. In Figs. 2 and 3, we presented simulations using synthesized random *T* matrices that were real and positive. This simulated the compression process if light coupled to each of the input waveguides was effectively incoherent. For example, a frequency comb or other multi-wavelength source could be used to couple light at different wavelengths into each waveguide (to be more quantitative, light in each waveguide should be separated in frequency by at least ~10X the detector bandwidth to minimize interference effects)[33]. In this case, the speckle patterns formed by light from each waveguide would sum incoherently on the detectors and the compression process can be

modeled using a random *T* matrix that is real-valued and non-negative. The second case we considered is using a complex-valued field *T* matrix, in which each element in *T* was assigned a random amplitude and phase. In this case, the compressed image was obtained as the square-law detector response: $O = \left(T\sqrt{I}\right)\left(T\sqrt{I}\right)_*$. This case simulates the effect of coupling a single, coherent laser to all the input waveguides at once such that the measured speckle pattern is formed by interference between light from each waveguide.

To evaluate the trade-off between real and complex transforms, we evaluated the reconstructed image quality at varying noise levels. In general, noise could be introduced in the original image formation process (e.g., due to low-light levels or imperfections in the imaging optics), through the camera opto-electronic conversion process (e.g., due to pixel non-linearity or the limited bit depth of the camera pixels), or through the optical compression process described in this work (e.g., due to laser intensity noise, environmental variations in the *T* matrix, or simply shot noise at the detection stage). To simulate the effect of noise on the reconstruction of the compressed images, we numerically added gaussian white noise to the compressed images. Figure 4a, d shows the same test image evaluated in Fig. 2, compressed by a factor of 8X (compression ratio 1:8) using either a real-or complex-valued 8 × 8 pixel *T* matrix. In this case, gaussian white noise with an amplitude equal to 2% of the average signal level in the image (corresponding to an SNR of 50) was added to each compressed image. The reconstructed images using the real and complex *T*

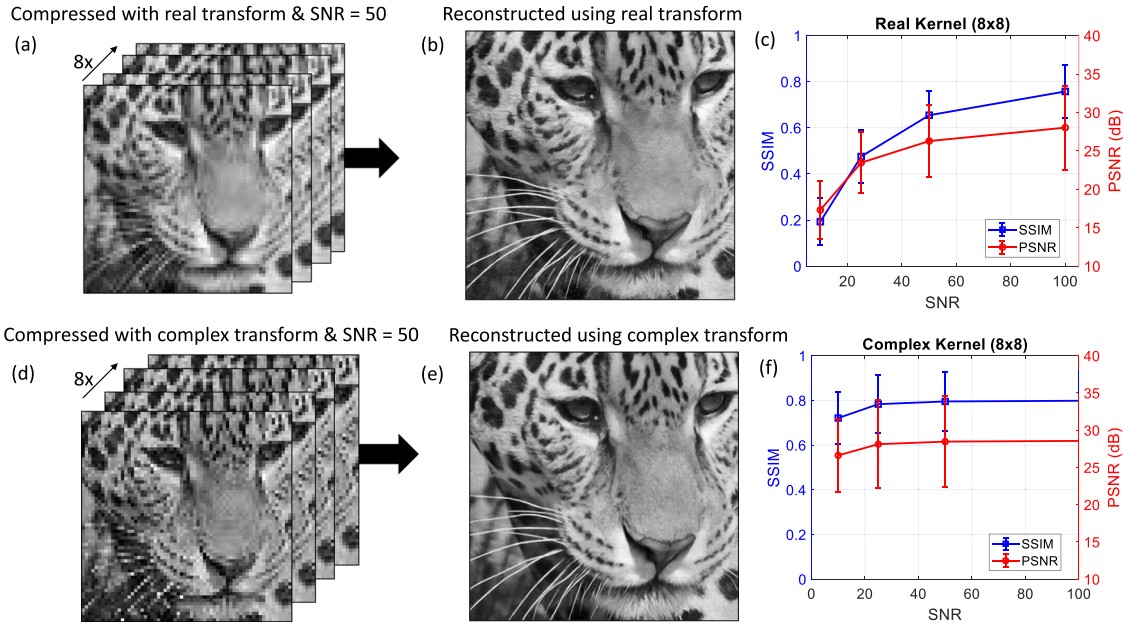

**Fig. 4 | Effect of kernel type (real vs complex) on image compression and reconstruction using images from the DIV2K and Flickr2K dataset[31,32]. a** 64 × 64-pixel images that were compressed using an 8×8 real kernel with 2 % gaussian white noise (SNR = 50). The original image was 512 × 512 pixels and is shown in Fig. 2(a). **b** Reconstructed image (512 × 512-pixel) from the compressed images shown in (a). **c** Mean PSNR (red) and SSIM (blue) of the reconstructed images from the test data as a function of SNR. The error bars correspond to 1 standard deviation. **d** 64 × 64-pixel images that were compressed using an 8 × 8 complex kernel with 2 % gaussian white noise (SNR = 50). The origin al image was 512 × 512 pixels and is shown in Fig. 2(a). **e** Reconstructed image (512 × 512-pixel) from the compressed images shown in (**d**). **f** Mean PSNR (red) and SSIM (blue) of the reconstructed images from the test data as a function SNR. The error bars correspond to 1 standard deviation.

matrices are shown in Fig. 4b, e. At 2 % noise level (SNR = 50), the reconstructed images are only marginally worse than the reconstructed image without noise shown in Fig. 2c (PSNR = 25.1 dB with noise vs PSNR = 26.9 dB without noise for the case of a real transform). This measurement confirms that the autoencoder framework is relatively resilient to noise, which is consistent with prior applications of autoencoders for denoising tasks. This resilience could also enable the system to forego the energy-intensive image conditioning by encoding raw image data and relying on the back-end neural network to compensate for noise due to effects such as pixel non-uniformity. In Fig. 4c, f, we present the average PSNR and SSIM for reconstructed test images that were compressed using either real valued or complex valued $T$ matrices, as a function of the SNR of the compressed images. These simulations showed that at relatively high SNR ( > 50), the real and complex valued $T$ matrices provided comparable performance. However, at lower SNR, the complex valued $T$ matrices provided more robust image compression due to the higher contrast in the compressed images obtained using a complex $T$ matrix.

### Experimental image compression and denoising
Next, we performed experiments to validate the following key predictions:

(1)  To confirm that our proposed approach (using an analog photonics-based fixed linear random matrix for compression and a non-linear neural network for de-compression) could provide comparable quality image compression to JPEG (which uses an image-dependent compression scheme which is far more energy and time-intensive).

(2)  To confirm that this technique could be used for both de-noising and compression.

For experimental validation of our image processing approach, we fabricated a prototype device on a silicon photonics platform. The experimental device included $N$=16 single-mode input waveguides

connected to the scattering layer through a multimode waveguide. The multimode waveguide region allowed light from each single-mode waveguide to spread out along the transverse axis before reaching the random scattering layer. This ensured that we obtained a uniformly distributed random transmission matrix without requiring an excessively long random scattering medium, which would introduce excess loss through out-of-plane scattering. To illustrate the impact of the multimode waveguide, we performed full-wave numerical simulations of single-mode waveguides either connected directly to the scattering layer or connected through an intermediate multimode waveguide region. In the first case, shown in Fig. 5a, the scattering layer was not thick enough for light to fully diffuse along the transverse axis, resulting in a high concentration of transmitted light near the position of the input waveguide. In terms of a $T$ matrix, this resulted in stronger coefficients along the diagonal rather than the desired, uniformly distributed random matrix. We then simulated the effect of adding a 32 μm multimode waveguide between the single-mode input waveguide and the scattering layer. As shown in Fig. 5(b), the multimode waveguide allowed light from the single-mode waveguide to extend across the scattering layer, resulting in a transmitted speckle pattern that was uniformly distributed.

The device was fabricated using a standard silicon-on-insulator wafer with a 250 nm thick silicon layer. The fabricated device consisted of 16 single-mode input waveguides connecting the device to the edge of the chip. The waveguides were 450 nm wide and separated by 3 μm (corresponding to a spacing of ~2λ at a wavelength of 1550 nm to minimize evanescent coupling). All 16 waveguides were connected to a 55.2 μm wide, 120 μm long multimode waveguide region, followed by a 30 μm long scattering region. The scattering region consisted of randomly placed 50 nm radius cylinders with a 3% filling fraction etched in the silicon waveguiding layer. The scattering layer parameters were empirically optimized to achieve a transmission of ~30% [see Supplementary Information: Section S5. Additional Experimental Characterization for experimental results]. To minimize leakage of light at the

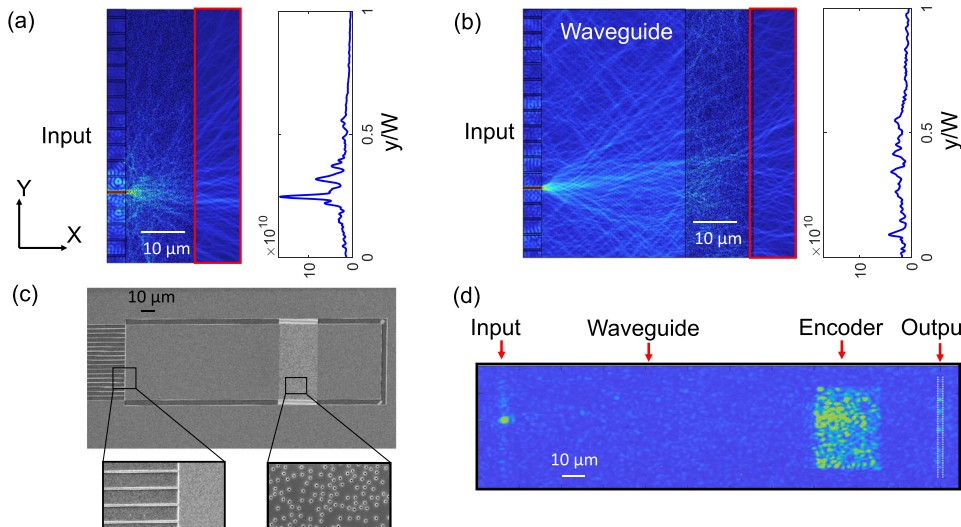

**Fig. 5 | Numerical simulations and experimental characterization. a, b** Full-wave frequency domain numerical simulations of the optical encoder with no (a) and a 32 μm multimode waveguide (**b**) in front of a 15 μm scattering region. Addition of the multimode waveguide region leads to spreading of the input light laterally along Y resulting in a random transmission matrix. The intensity along Y (normalized by width W) is plotted which is obtained by integrating the intensity along X within the region shown by the red boxes (**c**) Scanning electron micrograph of the fabricated silicon-photonics based all-optical image encoder. **d** An example of a typical experimental measurement performed for characterization of the transmission matrix of the encoder.

edges of the scattering layer, we added a full band-gap photonic crystal layer on the sides of the scattering layer[34–36]. We experimentally confirmed that transmission through the device was ~30%, as described in the Supplementary Information [Supplementary Information: Section S5. Additional Experimental Characterization]. Since this initial prototype did not include integrated photodetectors, we etched a ridge in the silicon waveguiding layer after the scattering region. This allowed us to record the light scattered out-of-plane from this ridge to measure the optical power which would be recorded if detectors were integrated in the device. Scanning electron microscope images of the fabricated device are shown in Fig. 5c.

We note here that the design goals of the random scattering medium for this application are significantly different compared with prior applications of on-chip scattering media such as the speckle spectrometer reported in Ref. 36. While a speckle spectrometer relies on obtaining distinct random projections for different input wavelengths, necessitating a relatively large, strongly-scattering region, here we desire a scattering medium with a broadband response (to minimize temperature dependence), but distinct, uniformly distributed random projections for different spatial modes. The compression device should also be designed to minimize the footprint and loss while still providing the type of fully distributed random transmission matrix required for high-quality image compression. As shown in Fig. 5 a, b, we found that a multimode waveguide region followed by a short scattering region allowed us to achieve this combination since it allowed each spatial input to overlap before reaching the scattering medium. This provided a uniformly distributed transmission matrix without requiring a large scattering region which would add significant loss.

To test the device, we first measured the *T* matrix by coupling an input laser operating at a wavelength of 1550 nm into each single mode waveguide and recording the speckle pattern scattered from the detection ridge after the scattering layer using an optical microscope setup. A typical image recorded using the optical setup is shown in Fig. 5d.

In order to account for experimental noise in the image compression and recovery process, we recorded two sequential *T* matrices, as shown in Fig. 6a, b. The *T* matrix was highly repeatable, as revealed in Fig. 6c, which shows the difference between the two

measurements. A histogram of the difference in the matrices, shown in Fig. 6d, indicates a gaussian-like random noise with amplitude ~1% of the average signal value (corresponding to a measurement SNR ~ 100). As shown in Fig. 4, at this SNR, both real and complex transformations provide similar results in terms of image reconstruction. This implies that we can use the experimentally measured intensity transmission matrix for image compression.

We note here that the experimental noise is due to noise generated by the laser and the electronics as expected for a real application. The photonic encoder, on the other hand, is extremely stable and provides a highly repeatable response. To confirm this, we monitored the transmission matrix for 60 h and the results are shown in the Supplementary Information [Supplementary Information: Section S5. Additional Experimental Characterization]. We found that the device is very stable with negligible fluctuations for the span of 60 h without requiring active temperature stabilization. This level of stability for an integrated photonic device is not surprising given the short lifetime of light passing through the scattering region, corresponding to a low effective quality factor with minimal temperature-dependence. Regarding temperature-dependence, based on our previous work[36] and assuming a thermo-optic coefficient in Si of $dn/dT \approx 1.8 \times 10^{-4}\,K^{-1}$ [37], the generated speckle pattern at the output will stay correlated for temperature changes up to ± 4°K. This stability is one of the advantages of the compact scattering device structure used in this work. Moreover, as shown later, using our unique approach of combining image compression with denoising, some of the noise introduced by thermal fluctuations during image compression could potentially be suppressed by training the back-end image reconstruction neural network using data acquired at a range of temperatures.

To convert the raw measured transmission matrix into the *T* matrix used for compression, we selected 4 non-overlapping spatial regions along the output ridge shown in Fig. 5d. This corresponds to selecting 4 columns of the matrix shown in Fig. 6a. This updated *T* matrix had dimensions of 16 × 4, providing a compression factor of 1:4. We then used this experimental matrix to train the back-end neural network required to reconstruct the original image. Note that we included noise in the training process by adding gaussian noise with the same 1% variance measured experimentally. Finally, we compressed the test images in the DIV2K and Flickr2K dataset while again

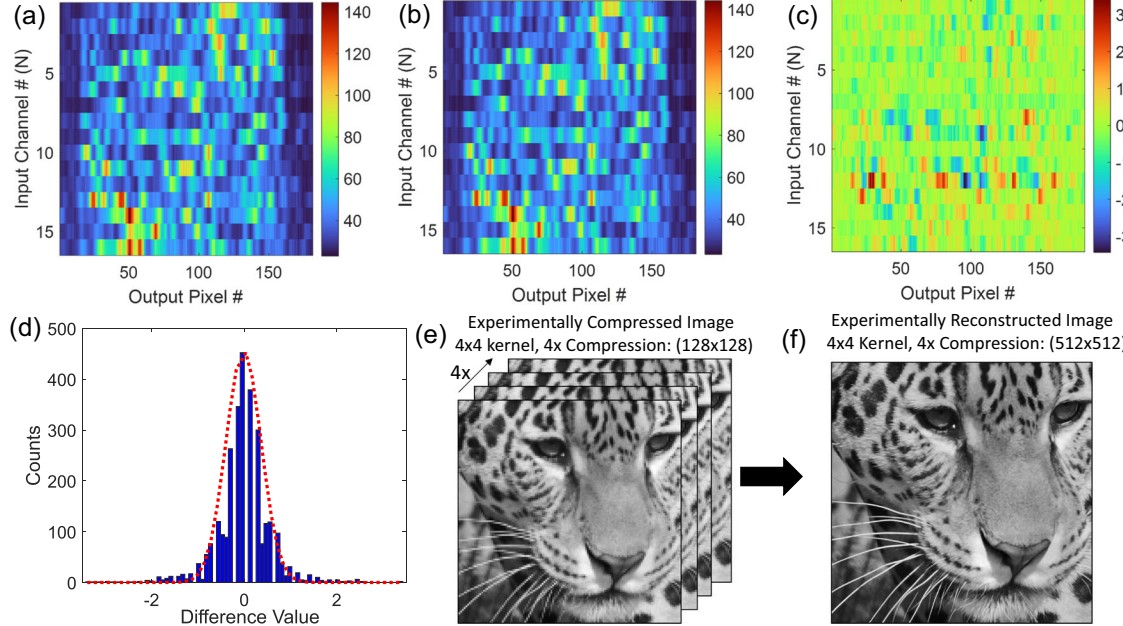

**Fig. 6 | Experimental demonstration of denoising and image compression using images from the DIV2K and Flickr2K dataset[31,32]. a, b** Experimental measurements of transmission matrices of the encoder are shown in Fig. 5(c). The two measurements shown in (**a**) and (**b**) are of the same device but measured at different times. The transmission matrices are extracted from a series of measurements similar to the one shown in Fig. 5d. The *x*-axis of the data corresponds to the pixel number at the output (image shown in Fig. 5d) and *y*-axis corresponds to the number of the input channels. **c** 2D plot showing the difference in magnitude of the different elements of the transmission matrices measured in (**a**) and (**b**). The difference in magnitude is on the order of ~1 %. **d** Histogram showing the distribution of the difference values measured in (**c**). The distribution corresponds to a gaussian distribution centered around 0. The red-dotted line shows the gaussian function fitted to the experimentally measured data. The red-dotted gaussian function was used as the noise source in the reconstruction algorithms. **e** An example of one of many compressed images taken from the DIV2K and Flickr2K dataset[31,32] where the compression was done using the experimentally measured transmission matrix. The original image is shown in Fig. 2(a) and compression ratio is 1:4 with each image being a 128 x 128 pixel image. **f** Reconstructed 512 x 512-pixel image from the compressed image shown in (**e**).

adding random noise with 1% variance. A typical compressed image using the experimentally measured *T* matrix is shown in Fig. 6e and the corresponding reconstructed image is shown in Fig. 6f. Excellent agreement between the original and reconstructed images can be seen with PSNR = 26.02 dB and SSIM = 0.91. In comparison, 1:4 JPEG compression of the same image and with similar SNR gives PSNR = 29.63 dB and SSIM = 0.83. We repeated this process for the entire set of test images and obtained an average PSNR of $26 \pm 4$ dB and SSIM of $0.9 \pm 0.07$. Additional examples of compressed and reconstructed images along with comparisons to JPEG compression are shown in the Supplementary Information [Supplementary Information: Section S1. Additional Experimental results: Compressed and reconstructed images and their statistics and Section S6. Comparison of Digital JPEG and Photonic Compression].

Finally, in addition to showing that this approach is robust to noise introduced in the analog photonic image compression step, this demonstration also illustrates how this technique could be used for image denoising. From the perspective of the back-end image reconstruction neural network, noise added during the original image acquisition process (e.g., due to pixel noise, non-uniform responsivity, or simply low light levels) is equivalent to noise added during the image compression step (as tested explicitly here using experimental data for noise). Thus, this work also highlights the potential for this technique to move the energy-intensive image conditioning and denoising steps to the back-end image reconstruction stage.

While this initial demonstration was performed at low speed because of the lack of integrated detectors and modulators, this approach is compatible with high-speed operation (e.g., >10 GHz). Our compression device can be considered as a photonic communication link where data is encoded on an optical carrier using the input modulators, transmitted through the scattering region (analogous to

transmission along a bus waveguide or through a fiber in a communications link) and is recorded on a high-speed photodetector. Since the loss through the scattering region of 30% is relatively low, we expect the compression device to operate with comparable SNR to photonic links operating at similar speeds. One potential complication is that noise will be added during the data encoding step (i.e., on the high-speed integrated modulators). However, our approach can allow us to compress and denoise simultaneously, and the image compression process works well even for SNR as low as 10 (as shown in Fig. 4f). In addition, we performed simulations where we calculated the compression quality as a function of noise added to the input image (simulating the effect of noise introduced by the modulators while encoding the inputs). The results are shown in the Supplementary Information (S4: Denoising Images), and they confirm that our approach is quite robust to noise introduced by the input modulators.

## Predicted energy consumption and operating speed for the photonic image processor

As described above, our encoding and compression technique can be reduced to a matrix multiplication operation. In order to compare the power consumption using our photonic approach with a traditional electronic scheme, we estimated the energy per multiply-accumulate (MAC) operation using both approaches. Electronic hardware accelerators have been thoroughly optimized to reduce the power consumption per MAC.

The power consumed by the photonic image processing engine includes contributions from the power consumed by the laser, the optical modulators, and the photodetectors. To estimate the required laser power, we first estimated the required detected power to provide sufficient signal-to-noise for accurate image compression. Assuming shot-noise limited detection, we can express the required optical

power reaching each photodetector as[38]:

$$P_{Rx} = 2^{2ENOB} q f_0 / \mathscr{R} \qquad (1)$$

where *ENOB* is the required effective number of bits, $q$ is the charge of a single electron ($1.6 \times 10^{-19}$ coulombs), $f_0$, is the operating frequency of the modulator (and the detector baud rate), and $\mathscr{R}$ is the responsivity of the photodetector in units of $A/W$. The *ENOB* can be related to the measurement SNR in dB as $SNR = 6.02 \times ENOB + 1.72$[38]. In the energy consumption calculations below, we assumed a required *ENOB* of 6, corresponding to a measurement SNR of 38 dB, which provides significant margin compared with the experimentally measured SNR of 17 dB. Based on the required power at the detector, we can work backwards to estimate the required laser power as

$$P_{laser} = \frac{N \times P_{Rx}}{T_{mod} T_{scatter}} \qquad (2)$$

where $N$ is the number of pixels in an image block, $T_{mod}$ is the transmission through the optical modulators, and $T_{scatter}$ is the transmission through the scattering medium. The electrical power required to drive the laser can then be written as $P_{laser}/\eta$, where $\eta$ is the wall-plug efficiency of the laser. The factor of $N$ in Eq. (2) implies that the multimode waveguide and scattering region support $N$ spatial modes (the minimum required to efficiently couple light from $N$ single-mode input waveguides) and each detector collects, on average, $1/N$ of the light transmitted through the scattering medium. In our preliminary experiment, presented above, a slightly larger multimode waveguide than required was used to simplify the experiment. As a result, the optical power was distributed over more than $N$ modes in our initial demonstration. In the future, adiabatically coupling the single-mode input waveguides into an $N$-mode multimode waveguide would optimize the power efficiency.

The power required by the optical modulators can be expressed as[39]

$$P_{Mod} = \frac{1}{2} C_{Mod} V_{pp}^2 f_0 \qquad (3)$$

where $C_{Mod}$ is the capacitance and $V_{pp}$ is the peak-to-peak driving voltage of the modulator. The power required by the photodetectors can be approximated as

$$P_{PD} \approx V_{bias} \mathscr{R} P_0 \qquad (4)$$

where $V_{bias}$ is the bias voltage of PN junction. The total electrical power consumed by the photonic image-processing engine can then be calculated as

$$P_{total} = P_{laser}/\eta + N \times P_{mod} + M \times P_{PD} \qquad (5)$$

Since the total number of MACs per second is $N \times M \times f_0$, the energy consumption per MAC is given by $P_{total}/(N \times M \times f_0)$. After substituting Eq. 1 into the expressions for $P_{laser}$ (Eq.2) and $P_{PD}$ (Eq. (4)), we see that the total energy consumption per MAC is independent of the modulation frequency.

To quantitatively compare the energy per MAC required by an optimized photonic processing engine with a conventional electronic GPU, we assumed typical specifications for the opto-electronic components. $C_{Mod}$ is usually on the order of 1fF, $V_{pp}$ is ~1 V, $V_{bias}$ is typically 3.3 V, and $\mathscr{R}$ is typically ~1 mA/mW at a wavelength of 1550 nm[40,41]. In addition, typical insertion loss for high-speed optical modulators is ~ 6.4 dB ($T_{mod} = 0.27$) and the wall-plug efficiency for distributed feedback lasers is $\eta = 0.2$[42,43]. The transmission through the experimental scattering medium is assumed to be $T_{scatter} = 0.2$ which also takes into account coupling efficiency to the integrated photodetectors. To be

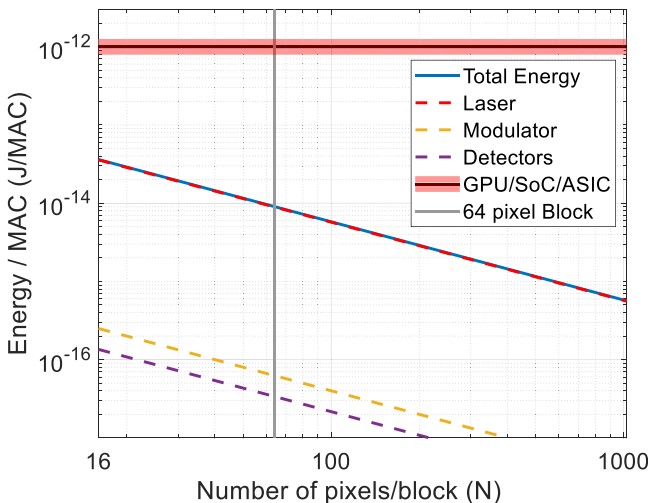

**Fig. 7 | Comparison of energy consumption for electronic and all-optical encoding approaches for image compression.** Comparison of energy consumption (given by Energy/MAC) as a function of number of inputs pixels $N$ for encoding via matrix multiplication using electronic approaches such as GPU, SoC, or ASIC-based (black solid line with a red band) and the presented optical approach using a photonic encoder with average transmission $T_{scatter} = 20\%$ (blue solid line). For the electronic approaches, typical energy efficiency of these digital accelerators is on the order of ~1 pJ/MAC and the red band shows the range of values for different accelerators. For the photonic approach, different color-coded lines correspond to the different components contributing to the total energy consumption such as the laser (dashed red line), modulators (dashed yellow lines), and detectors (dashed magenta line). The vertical gray line corresponds to energy consumption values for an $8 \times 8$ kernel or a 64-input pixel block. The energy consumption is dominated by the input laser and the overall energy efficiency of the presented optical approach improves as N or number of input pixels increases.

specific, our scattering medium provides a transmission of 30% as shown in the Supplementary Information [Supplementary Information: Section S5. Additional Experimental Characterization]. However, we assumed 20% overall transmission, which included ~67% transmission to the photodetectors (see Supplementary Information: Section S3. Integration of photonic encoder with silicon photonics and CMOS components for more details).

The estimated energy consumption per MAC as a function of the image block size $N$ is shown in Fig. 7. The energy required by the photonic image processor decreases rapidly with image block size. We also find that the majority of the power consumption is driven by the laser and $P_{laser}$ is 9.2 mW for a kernel size of $8 \times 8$ pixels and an ENOB of 6. While this power level is readily available from most commercial lasers operating in this wavelength regime (near 1550 nm), lower power consumption could be achieved if a lower *ENOB* was sufficient for a given application, which would enable a lower power laser (see Fig. 4c, f for an analysis of the trade-off between image reconstruction quality and the SNR of the compressed image). Nonetheless, we find that for an image block size of $8 \times 8$ pixels, the photonic image processor has the potential to provide 100X lower power consumption than a typical GPU. Although the photonic processor is even more efficient when using larger image blocks; this can degrade the image reconstruction, as shown in Fig. 2(d). In the future, alternative inverse-designed transforms could enable large pixel blocks without sacrificing image reconstruction fidelity.

We can also use this framework to estimate the energy consumption per pixel, which is calculated as $P_{total}/(N \times f_0)$. The energy per pixel is independent of both modulation frequency and the size of the pixel blocks, and for an *ENOB* of 6 and the parameters listed above,

can be as low as 72 fJ. This is dramatically lower than the ~0.1 $\mu$J used in existing image processing systems; however, the latter also includes the power required to operate the pixels and the analog-to-digital conversion process used to extract the signal recorded by each pixel. Nevertheless, since more than 50% of the energy consumed by standard electronic image processing systems is dedicated to image compression and conditioning, our opto-electronic approach can contribute significantly to reducing the overall energy consumption.

Finally, the device throughput, in terms of pixels/second, can be estimated as $N \times f_0$. Assuming an image block size of $8 \times 8$ ($N = 64$), this approach can process a Terapixel/second of image data using a clock speed of ~16 GHz, which is readily achieved using standard optical modulators and photodetectors[16–18]. While such compression would require significant temporal multiplexing to provide the compression engine with 64-pixel kernels at a rate of 16 GHz, fortunately, current on-board memory has the capability to store as much as 1 Gpixel of data in a buffer to feed the compression engine for real-time processing. Since the compression engine can process 1Tpixel/sec (or 1 Gpixel/msec), this is sufficient to keep up with the data rate acquired by a Gpixel camera operating at 1 kHz (well beyond current state-of-the-art). This speed is far beyond the compression speeds of conventional state-of-the-art digital electronics schemes such as JPEG compression which can compress at most around a Gigapixel/sec.

While the analysis presented above compares the energy consumption of our photonic approach to GPUs implementing the same random transform-based compression algorithm, another important comparison is the energy required for JPEG compression using a variety of processors that might be found on a digital camera or smart phone. To calculate the energy consumption for JPEG compression[19], first, we analyzed the number of operations required by the JPEG algorithm. The original JPEG compression algorithm performs a discrete cosine transform (DCT) followed by thresholding (to store the most significant basis functions) for every $8 \times 8$, non-overlapping kernel of pixels. To encode an image with $N \times N$ pixels, $N^2/64$ DCTs, each requiring $64 \times 64$ multiply and accumulate operations (MACs), are necessary, totaling $64N^2$ MACs. The more recent JPEG 2000 compression algorithm[44] decomposes the image into wavelet representations by successively passing it through 2D $n$-tap filters and applying $2 \times 2$ down sampling. For an image with $N \times N$ pixels undergoing $K$-level decompositions, the total number of MACs is $4n \sum_{j=0}^{K-1} N/2^j$. The computing complexity of both JPEG and JPEG 2000 scales with $O(N^2)$. This is the same computing complexity that our algorithm would require if you implemented the random projections using digital electronics rather than analog photonics.

Modern digital cameras or smart phones use an embedded system-on-chip (SoC) as the core processing unit. These SoCs typically include ARM cores for general-purpose computing, memory interface, and device control, along with specifically designed hardware codecs (e.g., JPEG/HEIF encoder in Canon DIGIC) or image-processing ASIC (e.g., Apple A16 Bionic) for JPEG compression and decoding. The power consumption of these codecs or ASICs can range from 0.5 to 20 pJ/MAC[45]. Because digital hardware codecs are designed for streaming pipelined, sequential inputs of image blocks, the total power consumption also grows as $O(N^2)$. Therefore, the power consumption per MAC remains constant regardless of the image size, sharing a similar scaling trend as GPUs. Moreover, the energy/MAC for a GPU (~1 pJ/MAC) is comparable to that of SOCs and ASICs and is plotted in Fig. 7.

Thus, to conclude, the power consumption for JPEG compression on a smartphone or digital camera is comparable to a GPU and exhibits the same scaling. The advantage of using a GPU for compression is that it might enable higher throughput by leveraging parallel processing. After this analysis, it is clear that the photonic compression engine still outperforms both in terms of speed and energy consumption with the potential for orders-of-magnitude lower power consumption than digital electronics solutions based on GPUs, ASICs or SOCs. Table 1 in

the Supplementary Information (Section S2. Energy consumption typical of mainstream electronic architectures) summarizes the energy consumption typical of mainstream electronic architectures, including desktop processors with the most efficient schemes reaching ~1 pJ/MAC.

## Discussion

In summary, we proposed a CMOS-compatible silicon photonics-based approach for large-scale image processing. Our approach performs image compression and de-noising using an auto-encoder framework in which the first layer of the network is implemented using analog photonics, while the back-end image reconstruction is implemented using digital electronics. Our approach enables image compression with comparable quality to standard digital compression techniques such as JPEG and image denoising quality comparable to state-of-the-art digital neural network based de-noising algorithms. In contrast to the prevailing image processing approach, which performs a large number of image conditioning tasks at the front end, our approach is designed to compress the raw image data and use the neural network back-end to both reconstruct the original image and perform the image denoising and conditioning operations.

In this work, we presented a combination of numerical simulations and experimental characterization of a proof-of-principle device to validate this approach. Using numerical simulations, we optimized the system design by evaluating the compression quality as a function of kernel size and transmission matrix type (positive, real-valued vs. complex-valued). These simulations confirmed that this scheme can provide comparable quality compression to the JPEG algorithm. We also evaluated the impact of noise on the image compression quality and demonstrated the ability of this scheme to perform de-noising. Finally, we analyzed the potential for this scheme to perform high-throughput processing with low power consumption. For example, by processing $8 \times 8$ blocks of pixels (i.e., 64 inputs) in parallel at ~16 GHz, this approach could process 1 Terapixel/s. This would enable large format image sensors such as Gigapixel cameras to operate at speeds up to 1 kHz or standard Megapixel imaging systems to operate at frame rates up to 1 MHz. From a power-consumption perspective, we found that an optimized photonic encoder would consume 100X less energy per MAC than a GPU. In addition to the numerical simulations and analysis, we experimentally characterized a passive silicon photonic prototype with 16 inputs designed to process $4 \times 4$ pixel kernels and demonstrated compression using a real-valued experimental transmission ($T$) matrix. This confirmed that the properties of the experimental transmission matrix enable image compression with similar quality to the digital JPEG algorithm and image denoising with similar quality to digital electronic neural network based denoising algorithms. This experimental characterization also confirmed that this approach is robust to fabrication imperfections, provided calibration is performed after fabrication. Future work will focus on integration of high-speed modulators and detectors and increasing the kernel size to support $8 \times 8$ pixel blocks.

While this work focused on compression and denoising of grayscale images, the general approach could be used to compress red-green-blue (RGB) images, hyperspectral, or time-series image data. Finally, this general scheme is amenable to a variety of imaging processing tasks other than compression including inference or classification. Again, the analog photonic transform could form the first layer of a neural network, accelerating the initial time and energy-intensive processing of high-dimensional image data, while relying on back-end digital electronics to complete the network. By tailoring this back-end neural network, the same photonic image processing engine could be applied to a variety of image processing and remote sensing applications.

Although we utilized a random scattering layer to generate the random encoding in this work, alternative approaches may be

advantageous in some applications. For example, multimode waveguides[46–48] or chaotic cavities[49] could also be used to perform random encoding for compression and have the potential for lower loss due to reduced out-of-plane scattering.

Finally, although we focused on using a linear encoding of the raw image data for compression, nonlinear, multi-layer encodings could potentially enable higher compression ratios. One trade-off is that non-linear encoding schemes are likely to require higher power consumption (due to optical attenuation and/or the power required to drive the non-linear process). Nonetheless, the potential for improved compression ratios and the ability to process larger kernel sizes at once makes this an intriguing approach which we hope to investigate in future studies.

## Methods

### Neural processing algorithm at the backend

All the images tested were from the DIV2K and Flickr2K dataset. A dataset of 4152 grayscale images was generated, each with a resolution of 512 × 512 pixels, through cropping and grayscale conversion. The dataset was divided into a training set of 3650 images and a validation set of 502 images. To study the image compression process, numerically generated random transmission matrices as well as experimentally measured transmission matrices were utilized as the encoding matrix. By using the original images as the ground truth and the compressed measurements as input, a convolutional neural network (CNN) was successfully trained to establish a correlation between the ground truth and the compressed images. The neural networks used were constructed based on Deep ResUnet[50] and ResUNet + +[51]. In order to investigate the impact of different compressive kernel sizes on the networks' ability to reconstruct compressed images, four different kernel sizes were explored (Fig. 2). The network architecture for the 4 × 4 kernel size consisted of 1 initial layer, 11 residual blocks, 2 down sampling layers, 4 up sampling layers, and 1 final convolutional layer. The residual block architecture was based on the residual neural network[52], and each block consisted of a Conv(3 × 3)-BN-LeakyReLU-Conv(3 × 3)- BN-[Conv(1 × 1)]-LeakyReLU block, with a Conv(1 × 1) layer added in the residual connection (marked with brackets). The downsampling (upsampling) rate in each downsampling (upsampling) layer was 2. The downsampling layer comprised of a Conv(3 × 3)-BN block, where the convolutional layer had padding =1 and stride = 2. The upsampling layer used ConvTranspose2d with kernel size 2 × 2 and stride=2. The initial layer was a Conv(3 × 3)-BN-LeakyReLU-Conv(3 × 3)-Conv(3 × 3) layer, which produced 64 feature maps, and the final convolutional layer had kernel size 1 × 1. For the networks with other kernel sizes, we added additional upsampling layers and residual blocks to maintain the number of layers and the size of the final output tensor. The network was trained with the mean squared error loss, Xavier initialization[53], Adam optimizer[54] with learning decay rate 0.1 per 400 epochs, and initial learning rate 0.001 for 800 epochs in PyTorch. The training was performed on four Nvidia V100S GPUs with a batch size of 64.

### Full-wave electromagnetic simulations

The simulated fields shown in Fig. 5a, b were obtained using the finite element method, (COMSOL with the Electromagnetic Waves, Frequency Domain interface) in a 2D geometry. The length of the simulated devices was scaled down to reduce the computational time and consisted of a scattering region 15 μm in length followed by a 10 μm long output port. Perfect conductors were substituted for the photonic crystal reflectors on the top and bottom sides of the scattering region. The scattering region contained a randomly generated set of 6180 holes with radius = 50 nm. Perfectly matched layer (PML) boundary conditions were used on the boundaries of the observation region. Ports were used to excite the input waveguides (width 0.45 um) which had a pitch of 3.45 um with 16 total inputs in all giving a total

device width of 55.2 μm (similar to fabricated structure). The pre-scattering-region length corresponding to the multimode waveguide region between the single mode waveguide inputs and the scattering region was varied between 50 nm and 32 μm and had a 4 μm air buffer on the side edges before absorbing PML boundaries. The effective material index of the silicon domains that we used for the 2D simulations was determined from 3D simulations and it was found to be $n_e = 2.83$. The effective material index of the air domains was $n_e = 1$. For the simulation, a triangular mesh was used with 1,256,090 elements with an average element quality of 0.92. A boundary mode analysis step was performed for each input waveguide port and the model was simulated at $\lambda = 1550$ nm. For analyzing the electromagnetic fields in the output region, the fields were exported on a regular grid with 100 nm steps in $X$ and $Y$.

### Sample fabrication

The silicon-photonics image encoder was fabricated using commercially available silicon-on-insulator (SOI) wafers. The wafers consisted of 250 nm silicon on top of a 3 μm buried oxide. The encoder was fabricated using a positive tone ZEP resist followed by electron beam lithography and inductively-coupled plasma reactive ion etching. The encoder consisted of $N = 16$ input single-mode waveguides, and the width of each waveguide was 450 nm. At the output, a ridge was fabricated to scatter the light out-of-plane, which was then measured to determine the transmission matrix of the encoder. The input waveguides were separated by 3.45 μm to minimize cross-coupling. The input waveguides were adiabatically tapered out towards the edge of the chip to increase the spacing between them to 10 μm, which was ~10X the size of the focused laser spot used for coupling the input light. This ensured that only one input waveguide was excited at a time during the measurement of the transmission matrix.

### Experimental measurement

To measure the transmission matrix of our encoder (Fig. 5), we used an aspheric lens to couple continuous wave (CW) laser light at $\lambda = 1550$ nm to the chip. The lens was used to focus the laser beam to a spot of diameter ~1.5 μm at the edge of the waveguides. To measure the transmission matrix, the focused laser spot was scanned to couple to each input waveguide while the transmitted speckle pattern was recorded from above using a long-working distance objective (50x, NA = 0.7) and an InGaAs camera (Xenics, Cheetah). All the images acquired during the measurement were then processed to determine the transmission matrix which we used for encoding the images. To quantify the experimental noise, the measurements were repeated, and the difference in magnitude of the elements of the transmission matrix was used as the experimental noise present in encoding the images.

## Data availability

The data that support the findings of this study are available from the corresponding authors upon request.

## Code availability

The codes and datasets in this study have been deposited in the Zenodo database under Creative Commons Attribution 4.0 International Public License at https://doi.org/10.5281/zenodo.10819458.

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

## Acknowledgements

This work was performed in part at the Center for Integrated Nanotechnologies, an Office of Science User Facility operated by the U.S. Department of Energy (DOE) Office of Science. This work was supported by the Laboratory Directed Research and Development program at Sandia National Laboratories, a multimission laboratory managed and operated by National Technology and Engineering Solutions of Sandia, LLC, a wholly owned subsidiary of Honeywell International, Inc., for the U.S. Department of Energy's National Nuclear Security Administration under contract DE-NA-003525. This paper describes objective technical results and analysis. Any subjective views or opinions that might be expressed in the paper do not necessarily represent the views of the U.S. Department of Energy or the United States Government.

## Author contributions

R.S., D.B., and B.R. conceived the idea. R.S. supervised the project. X.W., J.S., and D.B. developed the image processing and reconstruction algorithms. B.R. performed the optical characterization. N.K. and R.S. performed the electromagnetic simulations. R.S. and C.L. fabricated the samples. Z.Z. and S.P. performed the energy analysis. All the authors discussed the results and contributed to the writing and editing of the manuscript.

## Competing interests

The authors declare no competing interests.
