## [Peer Review File · Nature Communications]

Integrated Photonic Encoder for Low Power and High-Speed Image ProcessingREVIEWER COMMENTS

Reviewer #1 (Remarks to the Author):

In the manuscript, Wang et.al proposed an integrated photonic encoder aiming at fast and energy-efficient image processing, especially compression. The topic is very interesting and they have shown quite promising results. I definitely encourage the efforts of the authors in exploring optical compression on chips but I can hardly recommend this manuscript to be published in Nature Communications at the current stage until the authors address the following concerns and misleading parts.

Solid benchmarking

The authors claimed that 'We experimentally demonstrate image compression with a ratio of 4:1 and develop a back-end neural network capable of reconstructing the original images with an average peak signal-to-noise ratio (PSNR) ~ 25 dB and structural similarity index measure (SSIM) ~ 0.9 , comparable to common electronic software-based lossy compression schemes such as JPEG [10]'. However, the precise SSIM and PSNR with JPEG for the same image and compression ratio are not given in the experimental parts. Therefore, it is quite misleading when termed like this. I strongly suggest a direct clear comparison is inevitable before evaluations on the experimental performance of the proposed system can be drawn.

The claimed 'bottleneck'

The authors claimed in the introduction that 'the power consumption associated with electronic digital processing is now the bottleneck limiting image data acquisition rates'. Whereas, no references are cited for such an important conclusion, as well as the claimed main motivation of this work. I also suspect that transmission throughput could actually be one of the bottlenecks of large-scale data acquisition.

Performance with a high compression ratio

As the authors demonstrated in the simulation part, a small kernel (8x8) provides a limited compression ratio, while a larger kernel leads to a rapid decrease in reconstruction performance. I suspect whether this system could perform optical compression with a larger compression ratio and competitive performance, which already has been reported with all-optical encoders (DOI:10.1126/sciadv.adf8437 and DOI: 10.1126/sciadv.abd7690).

The term of auto-encoder neural network framework

The authors defined the proposed system with a random encoder and an electronic decoder as 'relies on an auto-encoder neural network framework'. Considering the employment of a random encoder, compressing sensing would be a more appropriate description. The authors themselves also cited references about compressed sensing instead of autoencoder in this part. And as a compressing sensing method, the authors should compare the experimental performance of SSIM and PSNR with more extensive compressing sensing methods.

Reviewer #2 (Remarks to the Author):

The authors propose an imaging system consisting of an optical frontend and a digital backend. The

optical frontend is a silicon-photonics processor that compresses incoming optical signals (the image) via a random linear transformation. These compressed analog optical signals are then converted to digital signals that are fed into a neural network (the digital backend) to reconstruct the original image. The authors experimentally see reasonably high-quality reconstruction with compression ratios of 4:1.

I believe this work will be of interest to the optical neural networks community and in principle recommend publication in Nature Communications. However, before I can concretely recommend publication, I believe the authors should address the following comments. I especially think it is important for them to appropriately reframe their claims in the title, introduction and abstract to accurately reflect the experimentally demonstrated evidence.

Major comments:

- It is a little unclear to me if the authors are proposing to replace a camera, or the signal processing immediately after image data acquisition, especially when they mention applications in remote sensing. If it is the former, I would like the authors to comment on the prospect of using incoherent light inputs instead of coherent light inputs to their encoder. If it is the latter, I would like the authors to comment on how they would acquire the natural scene image data to modulate the lasers in their encoder.
- If I have understood the work correctly, the authors only have 16 inputs to their encoder. This could reasonably be scaled up to $O(100)$, but probably not larger, given the generally accepted limits to input dimension for silicon photonics. There seems to be a large amount of time multiplexing that needs to be done if Terapixel images are the inputs, which could significantly slow down the image acquisition rate. While Terapixel/s throughput might be possible, that does not immediately translate to frame rate when the image resolution is high. Could the authors comment on this?
- I believe a lot of relevant work that has explored the prospects and advantages of optical encoders (not necessarily for image reconstruction) has not been cited in this manuscript. For example:
 - o Spectrally encoded single-pixel machine vision using diffractive networks
 - o An on-chip photonic deep neural network for image classification
 - o Single-shot hyperspectral-depth imaging with learned diffractive optics
- The authors compare the energy performance of their photonic encoder to a GPU. I do not think that is a fair comparison to state-of-the-art cameras since most cameras/low-level image processing algorithms do not use GPUs to compress their data. I believe a fair comparison would be to estimate how much energy a digital camera/smart-phone camera uses to compress an image to JPEG.
- The emphasis on Terapixel in the title and abstract is somewhat misleading, since the experimental demonstrations do not show performance on Terapixel images. I believe the title especially, and to a large extent also the abstract, should focus on what was experimentally demonstrated, and then the Discussion section of the paper is the appropriate place to speculate about the future possibilities of the work.

Minor Comments:

- While the authors have accounted for noise during compression, I do not believe they have

accounted for noise in the original image data. Since the authors are pitching their device as the front-end of an image acquisition pipeline, I would like to see, at least in simulation, how well their denoising auto-encoder structure works if the input image to the photonic encoder has a low SNR (as could well be the case with natural lighting conditions).

- It is well known in the deep-learning community that adding nonlinearity, and hence depth, to the encoder can allow for higher compression ratios (see this paper). Could the authors comment on the prospect of adding nonlinearity to their proposed encoder?

Reviewer #3 (Remarks to the Author):

This manuscript presents a conceptual discussion, theoretical analysis, and preliminary experimental investigation of an analog photonic high-throughput image compression system based on random multiplexing in a silicon photonic device and neural network based image reconstruction. The essential idea is to leverage an autoencoder framework wherein the compressed image is generated at a layer of reduced dimension in the latent space. The initial layer of the autoencoder is implemented in a physical photonic chip with random coupling between nodes such that the reduced dimension latent space is recorded on photodetectors at the output of this chip. With sufficient characterization and robustness of the encoding matrix the images can then be recovered using a trained neural network. The authors emphasize the utility of a local transform for this process theoretically showing promising results for an 8x8 pixel kernel and 8x compression. Experimentally, the authors investigate a 4x4 kernel and 4x compression in the prototype device. I find this work to be highly interesting and well evaluated from a theoretical standpoint. However, I find the experimental validation to be very limited and lacking in rigor, which makes it difficult to evaluate the experimental feasibility of the approach based on the presented results. I have the following suggestions for the authors to improve the manuscript.

1) My largest concern with this publication is that most of the work is theoretical predication that are not experimentally validated. The experimental work is superficial and only verifies the suitable randomness of the transmission matrix generated by the integrated photonic device. Other aspects are not experimentally verified such as the optical loss of the random multiplexing system, the ability to operate at high speed, the signal to noise ratio when operating at the proposed high speeds, the long term repeatability of the transmission matrix, and the operation of the system with multiple active inputs. In the context of experimental validation the only aspect that is validated is that the system provides sufficiently random input-output coupling that is repeatable over a (presumably) short interval of time (the time between measurements is not provided). Of all of the questions that could be experimentally validated this is probably the least surprising as based on the random scattering design and the wealth of previous work with similar designs (e.g. the referenced scattering speckle spectrometer work) it is expected that the system provides suitably random input-output coupling. Thus the results presented here are highly preliminary and mostly limited to the introduction of the concept of local random multiplexing using an integrated photonic device for the purposes of image compression and some theoretical verification of feasibility. Overall the paper would benefit from significantly much more experimental validation.

2) This concept itself is not very novel. While this system focuses on the use of neural networks for image recovery from a compressed representation of the image, the overall scheme is effectively a rebranding of that of compressed sensing using a photonic device for random multiplexing and implementation of the sensing matrix. Similar photonic devices have been developed in past work

for the purposes of compressive sensing (e.g. ref. [4] below and references therein, which is not referenced). Thus the concept of using such integrated photonic devices for image compression and compressed sensing presents limited novelty. Furthermore, there is extensive work on using neural networks rather than conventional l_1 minimization algorithms for compressed sensing and image recovery and thus this aspect also presents limited novelty.

3) Based on the limited experimental validation and the limited conceptual novelty, the main contribution of this work is the theoretical verification of feasibility at high data throughput. I do find this to be interesting, a well implemented theoretical verification, and publishable, however I find this contribution to be somewhat limited for publication in a journal such as Nature Communications.

4) Importantly, this work builds upon a body of work on integrated photonic devices that implement random multiplexing for a variety of purposes including neuromorphic computing, compressed sensing, and information security (e.g. refs [1]-[4] below). In this vein, while speckle spectrometers are referenced much of this other work is not referenced and should be added. The presented work should then be thoroughly discussed in the context of this previous work.

5) Relative to this previous work, the use of a random scattering layer implemented via defects in the multimode waveguide is a fairly simplistic and in many ways sub-optimal approach to implementing random multiplexing in an integrated photonic device. The main problem is that light scatters in all directions leading to out of plane losses and in plane losses from high incident angles at the multimode waveguide edges. While this is proposed to be mitigated to some extent for the in-plane cases through the incorporation of a photonic crystal boundary, there are better approaches demonstrated in past work. For example, the use of multimode waveguides with prescribed bends, chaotic cavities, or bidirectional evanescent coupling is highly promising as these devices achieve random multiplexing without scattering out of the device. The main benefit of the scattering approach is device length. A discussion of these alternative approaches and the relative benefits and downsides is warranted.

6) The theoretical performance of this approach is compared against JPEG in terms of PSNR and SSIM with comparable compression factors. While this is a useful comparison, another comparison that would be easier for a general audience to interpret would be to determine at what compression factor JPEG achieves the same PSNR/SSIM as comparing different PSNR/SSIM values is difficult for a general reader whereas compression factor is readily interpretable.

7) In the theoretical analysis the system loss and required laser power are considered in the calculations of SNR. However, the value of the required laser power is never provided. This is an important factor for determining feasibility and should be provided.

8) It does not appear that the coupling efficiency of the light in the waveguide to the integrated photodetectors is considered in the calculation. While photodetectors are not integrated here, what is the expected coupling efficiency based on the literature? Also it appears that these photodetectors will each couple to multiple spatial modes of the waveguiding structure. How many spatial modes per detector? How does this compare to the literature devices, which are likely designed to collect a light from a single mode structure?

9) Minor issue: The authors at one point state, "Unlike purely random data—which can be efficiently compressed using large random matrices—image data naturally includes spatial structure which is important to maintain." This statement is confusing and incorrect. Purely random data is not compressible as it is random and thus does not present sparsity in any domain of representation. I think that the authors intended to convey something else in this statement about the importance of designing a system that leveraged the spatial structure images and should reword this sentence to better convey their point.

Examples of references that should be added and discussed (more relevant references can be found

within these or referencing these):

- [1] K. Vandoorne, P. Mechet, T. Van Vaerenbergh, M. Fiers, G. Morthier, D. Verstraeten, B. Schrauwen, J. Dambre, and P. Bienstman, "Experimental demonstration of reservoir computing on a silicon photonics chip," *Nat. Commun.* 5, 3541 (2014).
- [2] B. Redding, S. F. Liew, Y. Bromberg, R. Sarma, and H. Cao, "Evanescently coupled multimode spiral spectrometer," *Optica* 3, 956-962 (2016).
- [3] B. C. Grubel, B. T. Bosworth, M. R. Kossey, H. Sun, A. B. Cooper, M. A. Foster, and A. C. Foster, "Silicon photonic physical unclonable function," *Opt. Express* 25, 12710-12721 (2017).
- [4] D. B. Borlaug, S. Estrella, C. T. Boone, G. A. Sefler, T. J. Shaw, A. Roy, L. Johansson, and G. C. Valley, "Photonic integrated circuit based compressive sensing radio frequency receiver using waveguide speckle," *Opt. Express* 29, 19222-19239 (2021).

Reviewer #1:

In the manuscript, Wang et.al proposed an integrated photonic encoder aiming at fast and energy-efficient image processing, especially compression. The topic is very interesting and they have shown quite promising results. I definitely encourage the efforts of the authors in exploring optical compression on chips but I can hardly recommend this manuscript to be published in Nature Communications at the current stage until the authors address the following concerns and misleading parts.

We thank the Reviewer for his/her insightful comments and questions which have helped us to significantly improve the manuscript. Below we address all the raised concerns.

Solid benchmarking

The authors claimed that ‘We experimentally demonstrate image compression with a ratio of 4:1 and develop a back-end neural network capable of reconstructing the original images with an average peak signal-to-noise ratio (PSNR) ~ 25 dB and structural similarity index measure (SSIM) ~ 0.9 , comparable to common electronic software-based lossy compression schemes such as JPEG [10]’. However, the precise SSIM and PSNR with JPEG for the same image and compression ratio are not given in the experimental parts. Therefore, it is quite misleading when termed like this. I strongly suggest a direct clear comparison is inevitable before evaluations on the experimental performance of the proposed system can be drawn.

Based on the Reviewer’s suggestion, for solid benchmarking, we have now compared all our results (including experimental results) to the digital JPEG image compression technique in the revised manuscript. We have added Figure 3 in the main manuscript, where we show direct comparison of our approach to JPEG compression as a function of compression ratio. In addition, in the experimental section as well as in Supplementary Information, we have now included the PSNR and SSIM of the images obtained using JPEG compression to compare with our experimental results. In both cases, we found that our approach provides PSNR/SSIM performance comparable to JPEG at low compression ratios (e.g., compression ratios $< 1:64$). More importantly, our approach actually allows us to achieve much higher compression ratios (e.g., compression ratios $> 1:64$) while providing acceptable PSNR > 20 dB up to high compression ratios $\sim 1:256$. This is particularly remarkable considering that our approach relies on “blind” compression using a fixed linear transformation as the front-end. In contrast, JPEG uses a power-hungry adaptive, non-linear compression scheme that is image dependent. Our work shows that training a neural network to reconstruct the image allows us to obtain comparable quality to JPEG despite using such a simplified front-end. Moreover, while the compression ratio in JPEG is image-dependent and limited to 1:64 for 512 x 512-pixel images, our approach can provide fixed and much higher compression ratios (variable compression ratios are particularly problematic for high-data rate image acquisition which must allocate variable-sized memory blocks as the compression ratio changes). Finally, by appropriately training the back-end neural network, our approach can also perform image conditioning and de-noising. This is not possible using digital JPEG image compression technique which only allows for compression.

We have now added a detailed discussion of this comparison in the section “Effect of kernel size and kernel type on image compressibility” of the revised manuscript.

The claimed ‘bottleneck’

The authors claimed in the introduction that ‘the power consumption associated with electronic digital processing is now the bottleneck limiting image data acquisition rates’. Whereas, no references are cited for such an important conclusion, as well as the claimed main motivation of this work. I also suspect that transmission throughput could actually be one of the bottlenecks of large-scale data acquisition.

We have now added four citations to the claim that power consumption is the bottleneck limiting image data acquisition rates:

- [1] Yan, X. et. al. "Compressive sampling for array cameras." *SIAM Journal on Imaging Sciences* **14** (1), 156-177 (2021).
- [2] Nichols, J. M. et. al., "Range performance of the DARPA AWARE wide field-of-view visible imager." *Appl. Opt.* **55**, 4478–4484 (2016).
- [3] Brady, D. J., Pang, W. B., Li, H., Ma, Z., Tao, Y., Cao, X., "Parallel cameras." *Optica* **5**, 127–137 (2018).
- [4] Wang, T. et. al., "Image sensing with multilayer nonlinear optical neural networks." *Nature Photonics* **17**, 408-415 (2023).

Refs. 1-3 show that in the high-resolution gigapixel imaging systems, image sensors operating at 30 fps draw 100 milliwatts/megapixel, whereas the image processing pipeline draws approximately 10X more power amounting to 200-1000 milliwatts/megapixel. This implies processing of terapixel worth of information would result in power consumption around 1 MW, which is prohibitive for many applications. Along with new citations, we have added this discussion in the revised manuscript. In addition, we have modified the text to: "*power consumption associated with electronic digital processing, along with limits on data transmission rate, and storage capacity are the major bottlenecks limiting image data acquisition rates [1-4].*"

However, we agree with the reviewer that data transmission, throughput, and storage is also a major bottleneck for large-scale image acquisition. Indeed, a major advantage of our approach is its potential to compress images at very high data rates (>1 Terapixel/s), which is far beyond the compression speeds of conventional state-of-the-art digital electronics schemes such as JPEG compression which can compress at most around a Gigapixel/sec. In the revised manuscript, we clarified our claim in the introduction, emphasizing that both energy consumption and data throughput are the primary bottlenecks to even higher data rate image acquisition.

Performance with a high compression ratio

As the authors demonstrated in the simulation part, a small kernel (8x8) provides a limited compression ratio, while a larger kernel leads to a rapid decrease in reconstruction performance. I suspect whether this system could perform optical compression with a larger compression ratio and competitive performance, which already has been reported with all-optical encoders (DOI:10.1126/sciadv.adf8437 and DOI: 10.1126/sciadv.abd7690).

We thank the reviewer for pointing out these references. We have added them to our citations. We do want to point out that our work is addressing a fundamentally different problem from the above-mentioned references. Our work is focused on low-power and high-speed compression of image data *after the image formation process on a focal plane array*. Most work on optical compression and compressed sensing, including these references, has focused on processing the scene data directly in the analog optical domain (i.e., at the image acquisition stage). While both schemes are valuable, we believe there are several important advantages for our approach, which has been relatively under-explored. First, our approach is compatible with conventional imaging optics and focal plane arrays, which are highly optimized. Second, our approach is immediately compatible with any image acquisition system, regardless of operating wavelength, camera resolution, front-end optics, frame rate, or application. There is no added insertion loss before the initial detection stage, enabling compatibility with low-light and high-speed imaging applications. Perhaps most significantly, our approach works with either ambient illumination imaging or active illumination, whereas the above references both require active illumination with a coherent source, which severely limits the application space. Lastly, by processing modest-sized kernels at a time, our

approach is inherently scalable and could process high-frame rate, limited region-of-interest image data, or Gpixel images at slower data rates. In contrast, these compressive sensing schemes are designed for a fixed (and relatively limited) imaging resolution.

In the revised manuscript, we clarified the use-case of our photonic compression engine and its potential advantages in the introduction and operating principle section.

Regarding the kernel size and compression ratio, we added simulations showing that our approach provides acceptable performance (PSNR > 20dB) for compression ratios as high as 1:256. We also tested our approach on the entire DIV2K and Flickr2K dataset, which contains a variety of highly dense and challenging images. It is difficult to make a direct comparison with the references above which reported compression results for the MNIST digit database (which consists of much lower resolution, sparser images) with no direct mention of PSNR/SSIM of reconstructed images. Moreover, these systems are targeting different applications, and our system is more directly competing with JPEG compression using digital electronics. As described above, we achieve comparable compression quality to JPEG with dramatically lower power consumption and higher throughput.

The term of auto-encoder neural network framework

The authors defined the proposed system with a random encoder and an electronic decoder as ‘relies on an auto-encoder neural network framework’. Considering the employment of a random encoder, compressing sensing would be a more appropriate description. The authors themselves also cited references about compressed sensing instead of autoencoder in this part. And as a compressing sensing method, the authors should compare the experimental performance of SSIM and PSNR with more extensive compressing sensing methods.

Although this work is inspired from compressive sensing theory, we do not perform conventional compressed sensing measurements. Traditional compressed sensing refers to compressed measurements in the measurement plane which in this case will be during image acquisition. However, in this work, what we are doing is something fundamentally different. We are compressing image data that has already been acquired by the imaging system and our work is more related to the problem of dimensionality reduction or data compression. Therefore, as the Reviewer pointed out before, we agree that JPEG image compression is the right method to compare our results for benchmarking. Based on the Reviewer’s suggestion, we have now compared all our results (including experimental results) to the digital JPEG image compression technique in the revised manuscript. We have added Figure 3 in which we directly compare our approach to JPEG compression. In addition, in the experimental part as well as in Supplementary Information, we have now included the PSNR and SSIM of the images obtained using JPEG compression. Finally, we call it an auto-encoder neural network framework because the back-end is a neural network and reconstruction is not based on an L2 norm minimization problem.

Finally, based on Reviewer’s suggestion, we modified the introduction and operating principle section of the revised manuscript to clarify this point. We also added a citation [Yuan, X., Haimi-Cohen, R., “Image compression based on compressive sensing: End-to-end comparison with JPEG.” *IEEE Transactions on Multimedia* **22(11)**, 2889-2904 (2020).] which compares CS schemes and JPEG compression for much smaller (256 x 256 pixel-size) image size and data size (8 images). As shown in that reference, it relies on utilizing a single-pixel camera or lensless camera for acquiring the measurements, which as described above, makes our work fundamentally different. In general, we want to point here that while compressed sensing has been employed to improve the performance of numerous imaging applications, its utilization for large and high-resolution imaging remains challenging in terms of the computation and acquisition effort involved.

Reviewer 2:

The authors propose an imaging system consisting of an optical frontend and a digital backend. The optical frontend is a silicon-photonics processor that compresses incoming optical signals (the image) via a random linear transformation. These compressed analog optical signals are then converted to digital signals that are fed into a neural network (the digital backend) to reconstruct the original image. The authors experimentally see reasonably high-quality reconstruction with compression ratios of 4:1.

I believe this work will be of interest to the optical neural networks community and in principle recommend publication in Nature Communications. However, before I can concretely recommend publication, I believe the authors should address the following comments. I especially think it is important for them to appropriately reframe their claims in the title, introduction and abstract to accurately reflect the experimentally demonstrated evidence.

We thank the Reviewer for his/her thorough review and insightful questions which have helped us to significantly improve the manuscript. Based on the Reviewer's suggestion, as described below, we have reframed our claims in the title, introduction, and abstract. In addition, below we address all the raised concerns.

It is a little unclear to me if the authors are proposing to replace a camera, or the signal processing immediately after image data acquisition, especially when they mention applications in remote sensing. If it is the former, I would like the authors to comment on the prospect of using incoherent light inputs instead of coherent light inputs to their encoder. If it is the latter, I would like the authors to comment on how they would acquire the natural scene image data to modulate the lasers in their encoder.

It is the latter—the goal here is to compress image data that is first acquired with a conventional camera and imaging system. Our system enables compression with low power and sufficiently high throughput to support the data-rates produced by the highest-speed cameras currently available. To clarify the role of our system, we can consider the entire image acquisition/compression process in 4 steps:

- (1) Conventional imaging optics form an image on the focal plane array of the camera.
- (2) Conventional focal plane array detectors convert the analog optical image to the electrical domain.

At this point we have two options:

(3a) Most commercially available cameras are designed to digitize the image data recorded on the focal plane array. Using this type of camera, we would then use a digital to analog converter (DAC) to drive the optical modulators on-chip, re-encoding the image information in the optical domain on an optical carrier.

(3b) Focal plane arrays are also commercially available which provide a direct analog output (NOII4SM6600A: <https://www.onsemi.com/pdf/datasheet/noii4sm6600a-d.pdf>). This analog output could be used to directly drive the optical modulators to re-encode the image information without an intermediate digitization step. A transimpedance amplifier (TIA) can directly convert the analog photocurrent to voltage to drive the re-encoding modulation, with appropriate amplification.

(4) The chip then performs high-speed, low power compression and the output from the detectors on-chip is digitized and stored for off-line image reconstruction.

Using focal plane arrays that provide an analog output (option 3b) has the potential to significantly reduce the overall power consumption and throughput by avoiding the intermediate analog to digital (ADC)/DAC steps. However, our underlying approach (the photonic chip performing compression) is compatible with both approaches, which is useful given the ubiquity of cameras with integrated ADCs.

In the revised manuscript, we clarified the role of our compression engine in the image acquisition/compression process in the operating principles section.

If I have understood the work correctly, the authors only have 16 inputs to their encoder. This could reasonably be scaled up to $O(100)$, but probably not larger, given the generally accepted limits to input dimension for silicon photonics. There seems to be a large amount of time multiplexing that needs to be done if Terapixel images are the inputs, which could significantly slow down the image acquisition rate. While Terapixel/s throughput might be possible, that does not immediately translate to frame rate when the image resolution is high. Could the authors comment on this?

The Reviewer is correct that the number of inputs can only be scaled up to $O(100)$. Assuming an image kernel size similar to JPEG, i.e. 8×8 (i.e. using 64 inputs) operating at 16 GHz, this approach could process $64 \times 16 = 1,024$ Gpixel/sec = 1.024 Tpixels/sec. This calculation was included at the end of the “Predicted energy consumption and operating speed” section. Of course, as the reviewer points out, this requires significant temporal multiplexing to provide the compression engine with 64-pixel kernels at a rate of 16 GHz. Fortunately, current on-board memory has the capability to store as much as 1 Gpixel of data in a buffer to feed the compression engine for real time processing. Since the compression engine can process 1Tpixel/sec (or 1 Gpixel/msec), this is sufficient to keep up with the data rate acquired by a Gpixel camera operating at 1 kHz (well beyond current state-of-the-art).

In the revised manuscript, we clarified the role and capabilities of on-board memory at the end of the “Predicted energy consumption and operating speed” section.

- I believe a lot of relevant work that has explored the prospects and advantages of optical encoders (not necessarily for image reconstruction) has not been cited in this manuscript. For example:
 - o Spectrally encoded single-pixel machine vision using diffractive networks
 - o An on-chip photonic deep neural network for image classification
 - o Single-shot hyperspectral-depth imaging with learned diffractive optics.

We thank the reviewer for pointing out these references. As mentioned earlier, these works targeted different applications, but we agree that they are relevant and help to illustrate the breadth of analog photonic data processing applications which have been explored in recent years. We have added these references in the introduction.

The authors compare the energy performance of their photonic encoder to a GPU. I do not think that is a fair comparison to state-of-the-art cameras since most cameras/low-level image processing algorithms do not use GPUs to compress their data. I believe a fair comparison would be to estimate how much energy a digital camera/smart-phone camera uses to compress an image to JPEG.

We thank the Reviewer for this good suggestion. Originally, we focused on our comparison with GPUs, which offer high throughput and might compete with our approach in high-data rate image acquisition applications. However, we agree with the reviewer and added a detailed comparison of the energy required for JPEG compression using a variety of processors that might be found on a digital camera or smart phone.

First, we analyzed the number of operations required by the JPEG algorithm. The original JPEG compression algorithm performs a discrete cosine transform (DCT) followed by thresholding (to store the most significant basis functions) for every 8×8 , non-overlapping kernel of pixels. To encode an image with $N \times N$ pixels, $N^2/64$ DCTs, each requiring 64×64 multiply and accumulate operations (MACs), are necessary, totaling $64N^2$ MACs. The more recent JPEG 2000 compression algorithm decomposes the image into wavelet representations by successively passing it through 2D n -tap filters and applying 2×2 down sampling. For an image with $N \times N$ pixels undergoing K -level decompositions, the total number of MACs is $4n \sum_{j=0}^{K-1} N/2^j$. The computing complexity of both JPEG and JPEG 2000 scales with $O(N^2)$. This is the same computing complexity that our algorithm would require if you implemented the random projections using digital electronics rather than analog photonics.

Modern digital cameras or smart phones use an embedded system-on-chip (SoC) as the core processing unit. These SoCs typically include ARM cores for general-purpose computing, memory interface, and device control, along with specifically designed hardware codecs (e.g., JPEG/HEIF encoder in Canon DIGIC) or image-processing ASIC (e.g., Apple A16 Bionic) for JPEG compression and decoding. The power consumption of these codecs or ASICs can range from 0.5 to 20 pJ/MAC. Because digital hardware codecs are designed for streaming pipelined, sequential inputs of image blocks, the total power consumption also grows as $O(N^2)$. Therefore, the power consumption per MAC remains constant regardless of the image size, sharing a similar scaling trend as GPUs. Moreover, the energy/MAC for a GPU (~ 1 pJ/MAC) is comparable to that of SOCs and ASICs.

Thus, the power consumption for JPEG compression on a smartphone or digital camera is comparable to a GPU and exhibits the same scaling. The advantage of using a GPU for compression is that it might enable higher throughput by leveraging parallel processing. After this analysis, the basic conclusion in the manuscript stands: the photonic compression engine has the potential for orders-of-magnitude lower power consumption than digital electronics solutions based on GPUs, ASICs or SOCs.

Based on the Reviewer’s suggestion, we have now included this discussion along with additional references in the revised manuscript (Section: Predicted energy consumption and operating speed for the photonic image processor). In addition, Figure 7 of the main manuscript and Table 1 in the Supplementary Information have been updated to include desktop processors, as well as ARM processors on embedded or mobile platforms.

Type	Name	Energy Efficiency
SoC	Apple A16 Bionic	0.4pJ/MAC
	JPEG hardware codec	0.5~20pJ/MAC
GPU	NVIDIA V100	4.6 pJ/MAC
	NVIDIA H100 SXM	0.7 pJ/MAC
ASIC	Google TPU v1	0.8 pJ/MAC
	Google TPU v4	1.2 pJ/MAC
FPGA	Xilinx Alveo U250	13.5 pJ/MAC
	Xilinx Versal VC2802	0.7 pJ/MAC

Desktop CPU	Intel i9-13900K	2.2nJ/MAC
	AMD Ryzen 9 7950X	2.3nJ/MAC

The emphasis on Terapixel in the title and abstract is somewhat misleading, since the experimental demonstrations do not show performance on Terapixel images. I believe the title especially, and to a large extent also the abstract, should focus on what was experimentally demonstrated, and then the Discussion section of the paper is the appropriate place to speculate about the future possibilities of the work.

Based on the Reviewer's suggestions, we have now modified the title to: "Integrated photonic encoder for low-power and high-speed image processing". We have also modified the abstract and now explicitly mention that we are referring to data rates of Terapixel/second and not Terapixel images.

While the authors have accounted for noise during compression, I do not believe they have accounted for noise in the original image data. Since the authors are pitching their device as the front-end of an image acquisition pipeline, I would like to see, at least in simulation, how well their denoising auto-encoder structure works if the input image to the photonic encoder has a low SNR (as could well be the case with natural lighting conditions).

We thank the Reviewer for this great suggestion. We have now performed simulations for the case when different levels of noise (quantified in terms of SNR) are added to the original image data. Our denoising auto-encoder structure performs quite well even when noise is present in the original image data. Even with SNR as low as 10, we achieve PSNR > 20 dB and SSIM > 0.6 for most test images, comparable to state-of-the-art image denoising techniques such as Wiener filtering, transform domain filtering methods, or convolutional neural network (CNN) based-methods (please see: Fan et. al, "Brief review of image denoising techniques", *Visual computing for industry, biomedicine, and art* 2, 7 (2019) for these methods and their performance). This confirmed that our compression and denoising scheme works well regardless of whether noise is introduced during the original image acquisition stage or during the compression process.

We have added Figures S4, S5, S6 and a new section in supplementary information (S4: Denoising Images) which shows some of the reconstructed noisy compressed test images and their corresponding PSNR and SSIM for different levels of SNR.

It is well known in the deep-learning community that adding nonlinearity, and hence depth, to the encoder can allow for higher compression ratios (see this paper). Could the authors comment on the prospect of adding nonlinearity to their proposed encoder?

This is a very insightful comment. We agree with the Reviewer that nonlinear, multi-layer encodings could potentially enable higher compression ratios and are actively investigating this possibility. One trade-off is that non-linear encoding schemes are likely to require higher power consumption (due to optical attenuation and/or the power required to drive the non-linear process). Nonetheless, the potential for improved compression ratios and the ability to process larger kernel sizes at once makes this an intriguing approach which we hope to present in a future paper.

For now, we have included a comment on this possibility in the Discussion section of the revised manuscript.

Reviewer 3:

This manuscript presents a conceptual discussion, theoretical analysis, and preliminary experimental investigation of an analog photonic high-throughput image compression system based on random multiplexing in a silicon photonic device and neural network based image reconstruction. The essential idea is to leverage an autoencoder framework wherein the compressed image is generated at a layer of reduced dimension in the latent space. The initial layer of the autoencoder is implemented in a physical photonic chip with random coupling between nodes such that the reduced dimension latent space is recorded on photodetectors at the output of this chip. With sufficient characterization and robustness of the encoding matrix the images can then be recovered using a trained neural network. The authors emphasize the utility of a local transform for this process theoretically showing promising results for an 8x8 pixel kernel and 8x compression. Experimentally, the authors investigate a 4x4 kernel and 4x compression in the prototype device. I find this work to be highly interesting and well evaluated from a theoretical standpoint. However, I find the experimental validation to be very limited and lacking in rigor, which makes it difficult to evaluate the experimental feasibility of the approach based on the presented results. I have the following suggestions for the authors to improve the manuscript.

We sincerely thank the Reviewer for his/her positive feedback and suggestions to improve the manuscript. In response to the Reviewer's suggestions, we performed additional experimental validation, including measurements of transmission loss and long-term stability. While high-speed operation will require foundry-fabricated devices with integrated high-speed modulators and detectors, we believe this is primarily an engineering challenge as high-speed photonic communication transceivers, which operate at similar speed and SNR, are now commercially available. We also added simulations evaluating the ability of this scheme to perform image denoising as well as a thorough comparison with JPEG based compression. Below we address all the suggestions of the Reviewer.

My largest concern with this publication is that most of the work is theoretical predications that are not experimentally validated. The experimental work is superficial and only verifies the suitable randomness of the transmission matrix generated by the integrated photonic device. Other aspects are not experimentally verified such as the optical loss of the random multiplexing system, the ability to operate at high speed, the signal to noise ratio when operating at the proposed high speeds, the long term repeatability of the transmission matrix, and the operation of the system with multiple active inputs. In the context of experimental validation the only aspect that is validated is that the system provides sufficiently random input-output coupling that is repeatable over a (presumably) short interval of time (the time between measurements is not provided). Of all of the questions that could be experimentally validated this is probably the least surprising as based on the random scattering design and the wealth of previous work with similar designs (e.g. the referenced scattering speckle spectrometer work) it is expected that the system provides suitably random input-output coupling. Thus the results presented here are highly preliminary and mostly limited to the introduction of the concept of local random multiplexing using an integrated photonic device for the purposes of image compression and some theoretical verification of feasibility. Overall the paper would benefit from significantly much more experimental validation.

Based on the Reviewer's suggestion, we have now included new experimental studies of (1) the optical loss of the random system and (2) the long-term repeatability of the transmission matrix. We have included these results in the Supplemental Information (S5: Additional Experimental Characterization) and in the revised manuscript. We found that the optical loss through the scattering structure is ~ 0.3 , consistent with our simulations and the transmission value used in our energy consumption analysis. For the stability measurement, we monitored the transmission matrix for 60 hours and the results are shown in Supplementary Information (S5: Additional Experimental Characterization). We found that the device is

very stable with negligible fluctuations for the span of 60 hours without requiring active temperature stabilization. We discontinued our measurements after 60 hours due to practical reasons. This level of stability for an integrated photonic device is not surprising given the short lifetime of light passing through the scattering region, corresponding to a low effective quality factor with minimal temperature-dependence. Regrading temperature-dependence, based on our previous work [Ref. 35] and assuming a thermo-optic coefficient in Si of $dn/dT \approx 1.8 \times 10^{-4} \text{ K}^{-1}$ [Ref.: Komma, J., Schwarz, C., Hofmann, G., Heinert, D., Nawrodt, R., “Thermo-optic coefficient of silicon at 1550 nm and cryogenic temperatures.” *Applied Physics Letters* **101**, 041905 (2012)], the generated speckle pattern at the output will stay correlated for temperature changes up to $\pm 4^\circ\text{K}$. This stability is one of the advantages of the compact scattering device structure used in this work. If needed, the thermal stability principle can be further improved using on-chip temperature stabilization techniques or by optimizing the lifetime of light passing through the scattering region by engineering the defect size and densities.

Moreover, using our unique approach of combining image compression with denoising, some of the noise introduced by thermal fluctuations during image compression could potentially be suppressed by training the back-end image reconstruction neural network using data acquired at a range of temperatures.

The remaining experimental validation suggested by the reviewer is testing at high-speed and measuring the SNR at high-speed (we have already measured the SNR at low-speed). While we agree that this will be an important validation (and we are currently working towards it), integrating active, high-speed modulators and photo-detectors requires foundry-level processing with a long-lead time and is beyond the scope of the current work.

However, we do not expect that operating at high-speed will significantly degrade the system performance. Photonic transceivers routinely operate at ~ 100 Gbps, which exceeds our required data throughput (e.g. <https://www.cisco.com/c/en/us/products/collateral/interfaces-modules/transceiver-modules/silicon-photonics-wp.html>). Our compression device can be considered as a photonic communication link where data is encoded on an optical carrier using the input modulators, transmitted through the scattering region (analogous to transmission along a bus waveguide or through a fiber in a communications link) and is recorded on a high-speed photodetector. Since the loss through the scattering region of 30% is relatively low, we expect the compression device to operate with comparable SNR to photonic links operating at similar speed. One potential complication is that noise will be added during the data encoding step (i.e. on the high-speed integrated modulators). However, our approach can allow us to compress and denoise simultaneously and the image compression process works well even for SNR as low as 10 (as shown in Fig. 4f). In addition, we performed simulations where we calculated the compression quality as a function of noise added to the input image (simulating the effect of noise introduced by the modulators while encoding the inputs). As shown in Figs. S4, S5, S6 and discussed in a new section in the supplementary information (S4: Denoising Images), our approach is quite robust to noise introduced by the input modulators.

Finally, we would like to point out that the experimental validation presented in this manuscript is not limited to a simple demonstration of a random transmission matrix. In particular, the design goals of the random scattering medium for this work were significantly different compared with prior applications of on-chip scattering media such as the speckle spectrometer reported in Ref. [35]. While a speckle spectrometer relied on obtaining distinct random projections for different input wavelengths, necessitating a relatively large, strongly-scattering region, here we desired a scattering medium with a broadband response (to minimize temperature dependence), but distinct, uniformly distributed random projections for different spatial modes. We also wanted a device with a minimal footprint and minimal loss, while still providing the type of fully distributed random transmission matrix required for high-quality image compression. As shown in Fig. 5 (a, b), we found that a multimode waveguide region followed by a short scattering region allowed us to achieve this combination, since it allowed each spatial input to overlap

before reaching the scattering medium. This provided a uniformly distributed transmission matrix without requiring a large scattering region which would add significant loss.

We added a discussion of these unique design requirements in the “Experimental image compression and denoising” section.

More generally, these experiments enabled two key demonstrations:

- (1) We experimentally confirmed that our proposed approach (using a fixed linear random matrix for compression and a non-linear neural network for de-compression) could provide comparable quality image compression to JPEG (which uses an image-dependent compression scheme which is far more energy and time-intensive).
- (2) We experimentally confirmed that that this technique could be used for both de-noising and compression, while JPEG is limited to performing compression.

Considering that the primary goal of this proof-of-concept work is to illustrate the potential for a photonic solution to address the energy consumption and data throughput bottlenecks limiting high data-rate image acquisition, we hope that these explanations along with the added experimental validation measurements satisfy the Reviewer’s concerns.

This concept itself is not very novel. While this system focuses on the use of neural networks for image recovery from a compressed representation of the image, the overall scheme is effectively a rebranding of that of compressed sensing using a photonic device for random multiplexing and implementation of the sensing matrix. Similar photonic devices have been developed in past work for the purposes of compressive sensing (e.g. ref. [4] below and references therein, which is not referenced). Thus the concept of using such integrated photonic devices for image compression and compressed sensing presents limited novelty. Furthermore, there is extensive work on using neural networks rather than conventional l_1 minimization algorithms for compressed sensing and image recovery and thus this aspect also presents limited novelty.

We respectfully disagree with the Reviewer on this point. To the best of our knowledge, this is the first work which shows that an analog hardware accelerator engine can be used to improve throughput and power consumption in an image acquisition pipeline *after image formation on the focal plane array*. This is in contrast to the significant body of work (as alluded to by the reviewer) focused on compressed sensing of the raw scene information. (Or, in the context of Ref. 4, compressing RF information in the analog domain at the RF acquisition stage).

There are several advantages to the type of back-end compression processor presented in this work. First, conventional imaging optics and focal plane arrays are highly optimized, and it is difficult to improve on their baseline performance, particularly in wavelength regimes where high-resolution focal plane arrays are available. This has limited the adoption of compressed sensing techniques designed to operate at the scene acquisition stage (many of which require active, coherent illumination). As a result, our approach does not attempt to alter the original image formation process. Instead, we designed an accelerator to address the two main bottlenecks limiting persistent, high-data-rate image acquisition: power consumption and compression speed. By positioning the accelerator after the image formation and initial optical-to-electrical conversion step, this scheme is compatible with any image acquisition system, regardless of operating wavelength, camera resolution, front-end optics, frame rate, or application.

In this work, we not only introduced the concept of using an analog compression accelerator engine, but showed that a passive, “blind” linear photonic compression engine followed by a neural-network decompression scheme can yield similar quality images to JPEG compression. This is remarkable since JPEG uses an adaptive, image-dependent compression scheme which is far more energy and time-intensive. We also showed that our scheme enables image denoising which is not possible using JPEG compression.

In the revised manuscript, we clarified the role of our image compression engine in the image acquisition pipeline and discussed its potential to enable the next-generation of high-data rate imaging systems.

Based on the limited experimental validation and the limited conceptual novelty, the main contribution of this work is the theoretical verification of feasibility at high data throughput. I do find this to be interesting, a well implemented theoretical verification, and publishable, however I find this contribution to be somewhat limited for publication in a journal such as Nature Communications.

We hope the above explanation addresses the Reviewer's concerns.

Importantly, this work builds upon a body of work on integrated photonic devices that implement random multiplexing for a variety of purposes including neuromorphic computing, compressed sensing, and information security (e.g. refs [1]-[4] below). In this vein, while speckle spectrometers are referenced much of this other work is not referenced and should be added. The presented work should then be thoroughly discussed in the context of this previous work.

We thank the Reviewer for pointing out these references. We have included them in the citations. In addition, as suggested by the Reviewer, we have now clarified the above-mentioned point (i.e., difference between our work and previous published literature) in the introduction of the revised manuscript to distinguish it from the previous published works.

Relative to this previous work, the use of a random scattering layer implemented via defects in the multimode waveguide is a fairly simplistic and in many ways sub-optimal approach to implementing random multiplexing in an integrated photonic device. The main problem is that light scatters in all directions leading to out of plane losses and in plane losses from high incident angles at the multimode waveguide edges. While this is proposed to be mitigated to some extent for the in-plane cases through the incorporation of a photonic crystal boundary, there are better approaches demonstrated in past work. For example, the use of multimode waveguides with prescribed bends, chaotic cavities, or bidirectional evanescent coupling is highly promising as these devices achieve random multiplexing without scattering out of the device. The main benefit of the scattering approach is device length. A discussion of these alternative approaches and the relative benefits and downsides is warranted.

This is a very interesting point. Yes, there may be alternative approaches which are advantageous in some applications, and these are worth investigating. In particular, multimode waveguides and chaotic cavities can be particularly interesting and will be a topic of our future studies. However, we want to point out that there are additional advantages of using a random scattering layer implemented via defects besides device length. For example, they are very robust to fabrication defects, being non-resonant they are broadband, and the spectral bandwidth of the transmission matrix (determined by the transport mean free path and loss) as well as the lifetime of light passing through the scattering region can be easily engineered by engineering the defect size and densities and the size of the scattering region. Most importantly, loss is quite low in the scattering layer: we performed updated experiments confirming that transmission loss is ~ 0.3 . Such low loss is possible since a very short scattering layer still provides sufficiently random projections for image compression. While trading off device size for lower loss (e.g. in a multimode waveguide) may be beneficial in some applications, we do not expect a dramatic change in performance given the modest loss in the scattering device used in this work.

We have included these discussions in the Discussion section of the revised manuscript.

The theoretical performance of this approach is compared against JPEG in terms of PSNR and SSIM with comparable compression factors. While this is a useful comparison, another comparison that would be

easier for a general audience to interpret would be to determine at what compression factor JPEG achieves the same PSNR/SSIM as comparing different PSNR/SSIM values is difficult for a general reader whereas compression factor is readily interpretable.

Based on the Reviewer suggestion, we have now compared all our results (including experimental results) to the digital JPEG image compression technique in the revised manuscript for different compression ratios. We have added Figure 3, which is a direct comparison of our approach to JPEG compression. In addition, in the experimental part as well as in Supplementary Information, we have now included the PSNR and SSIM of the images obtained using JPEG compression.

In the theoretical analysis the system loss and required laser power are considered in the calculations of SNR. However, the value of the required laser power is never provided. This is an important factor for determining feasibility and should be provided.

We have included this value in the revised manuscript. The required power is 9.2 mW for an ENOB of 6 and for an 8×8 kernel. This power level is readily available from most commercial lasers operating in this wavelength regime (near 1550 nm) and therefore the power required for desired SNR is not a problem.

It does not appear that the coupling efficiency of the light in the waveguide to the integrated photodetectors is considered in the calculation. While photodetectors are not integrated here, what is the expected coupling efficiency based on the literature? Also it appears that these photodetectors will each couple to multiple spatial modes of the waveguiding structure. How many spatial modes per detector? How does this compare to the literature devices, which are likely designed to collect a light from a single mode structure?

For the active devices, which will have integrated detectors, we intend to use the Germanium (Ge) photodiodes currently being developed by the Sandia National Laboratories CMOS compatible silicon photonics process. As shown by: DeRose et al., “Ultra compact 45 GHz CMOS compatible Germanium waveguide photodiode with low dark current”, Optics Express 19 (25) (2011), these photodiodes can be multimode with width ranging from 1.3 to 5.3 microns (i.e., can support from ~ 4 to ~ 20 modes at 1.55 microns). These photodiodes have a best-in-class 3 dB cutoff frequency of 45 GHz, responsivity of 0.8 A/W and dark current of 3 nA. In general, coupling efficiency in these waveguide integrated Ge detectors can be very high ($> 90\%$) without being limited by the intrinsic layer thickness of the device (see for example Ref. : Donghwan et al, “High performance, waveguide integrated Ge photodetectors”, Optics Express 15(7), 3916-3921 (2007)). In the next generation of data compression devices, we intend to further optimize coupling to the detectors by adiabatically tapering down the output silicon waveguides after the scattering region to match the number of modes supported by the Ge photodetector region.

Finally, we want to point out that we did account for coupling efficiency to the integrated photodetectors in our calculation. Our scattering medium provides a transmission of 30 % as shown in the Supplementary Information. However, we assumed 20 % overall transmission, which included $\sim 67\%$ transmission to the photodetectors.

We have included this discussion along with References in the Supplementary Information (Section S3. Integration of photonic encoder with silicon photonics and CMOS components) of the revised manuscript.

Minor issue: The authors at one point state, “Unlike purely random data—which can be efficiently compressed using large random matrices—image data naturally includes spatial structure which is important to maintain.” This statement is confusing and incorrect. Purely random data is not compressible as it is random and thus does not present sparsity in any domain of representation. I think that the authors intended to convey something else in this statement about the importance of designing a system that leveraged the spatial structure images and should reword this sentence to better convey their point.

We thank the Reviewer for pointing this out. We have modified the sentence to “*Unlike spatially uncorrelated and sparse data—which can be efficiently compressed using large random matrices—image data naturally includes spatial structure and spatial correlations which are important to maintain.*”

Examples of references that should be added and discussed (more relevant references can be found within these or referencing these):

[1] K. Vandoorne, P. Mechet, T. Van Vaerenbergh, M. Fiers, G. Morthier, D. Verstraeten, B. Schrauwen, J. Dambre, and P. Bienstman, “Experimental demonstration of reservoir computing on a silicon photonics chip,” *Nat. Commun.* 5, 3541 (2014).

[2] B. Redding, S. F. Liew, Y. Bromberg, R. Sarma, and H. Cao, “Evanescantly coupled multimode spiral spectrometer,” *Optica* 3, 956-962 (2016).

[3] B. C. Grubel, B. T. Bosworth, M. R. Kossey, H. Sun, A. B. Cooper, M. A. Foster, and A. C. Foster, “Silicon photonic physical unclonable function,” *Opt. Express* 25, 12710-12721 (2017).

[4] D. B. Borlaug, S. Estrella, C. T. Boone, G. A. Sefler, T. J. Shaw, A. Roy, L. Johansson, and G. C. Valley, “Photonic integrated circuit based compressive sensing radio frequency receiver using waveguide speckle,” *Opt. Express* 29, 19222-19239 (2021).

We thank the Reviewer for pointing out these references. We have included them in our citations.

REVIEWERS' COMMENTS

Reviewer #1 (Remarks to the Author):

The authors have fully addressed my concerns and I highly recommend the publication of this manuscript with their great efforts to show the effectiveness of the optical encoder at the chip scale with linear propagation. In the meantime, the recent work on the use of an optical encoder in 3D free space with an integrated optoelectronic chip should also be mentioned (Chen, Y., Nazhamaiti, M., Xu, H. et al. All-analog photoelectronic chip for high-speed vision tasks. Nature 623, 48–57 (2023).)

Reviewer #2 (Remarks to the Author):

I have reviewed the revised manuscript and I believe the authors have suitably addressed the referee concerns, so I recommend publication.

Feedback from reviewer #2 on reviewer #3 comments:
(Remarks to the Author)

I have reviewed the revised manuscript and the responses to Referee #3.

I think the authors have mostly addressed Referee #3's concerns, but I would recommend they do one more small edit to their manuscript:

Referee #3 makes the following remark: "My largest concern with this publication is that most of the work is theoretical predications that are not experimentally validated. The experimental work is superficial and only verifies the suitable randomness of the transmission matrix generated by the integrated photonic device..."

I agree that, even in the revised manuscript, there is a confusion about what the authors have theoretical proposed and simulated, versus what they have experimentally demonstrated.

To make this clearer, I would recommend that in the Discussion section, the authors explicitly add a paragraph that says something like the following: "We have proposed concept X that works in Y way. The key contribution here is Z. We have performed numerical simulations to validate the concept, and from these simulations predict that it is possible to engineer hardware with a 1024x1024 pixel array capable of performance [10 terapixels / second / Watt]. We have also performed proof-of-concept experiments with a toy-sized system (4x4 pixels), which achieves a performance [10 pixels / second / Watt], but validates that the randomness from C transmission is sufficient, and we have a reasonable expectation that if through foundry-scale efforts a state-of-the-art chip was produced, performance much closer to our theoretical/simulation predictions would be achieved."

Reviewer #1:

The authors have fully addressed my concerns and I highly recommend the publication of this manuscript with their great efforts to show the effectiveness of the optical encoder at the chip scale with linear propagation. In the meantime, the recent work on the use of an optical encoder in 3D free space with an integrated optoelectronic chip should also be mentioned (Chen, Y., Nazhamaiti, M., Xu, H. et al. All-analog photoelectronic chip for high-speed vision tasks. Nature 623, 48–57 (2023).)

We are happy that the Reviewer has recommended the publication of our manuscript. In the revised manuscript, we have added the citation mentioned above.

Reviewer #2:

I have reviewed the revised manuscript and I believe the authors have suitably addressed the referee concerns, so I recommend publication.

We thank the Reviewer for recommending publication and for his/her thorough review and insightful questions which have helped us to significantly improve the manuscript.

Feedback from reviewer #2 on reviewer #3 comments:

I have reviewed the revised manuscript and the responses to Referee #3.

I think the authors have mostly addressed Referee #3's concerns, but I would recommend they do one more small edit to their manuscript:

Referee #3 makes the following remark: “My largest concern with this publication is that most of the work is theoretical predications that are not experimentally validated. The experimental work is superficial and only verifies the suitable randomness of the transmission matrix generated by the integrated photonic device...”

I agree that, even in the revised manuscript, there is a confusion about what the authors have theoretical proposed and simulated, versus what they have experimentally demonstrated. To make this clearer, I would recommend that in the Discussion section, the authors explicitly add a paragraph that says something like the following: “We have proposed concept X that works in Y way. The key contribution here is Z. We have performed numerical simulations to validate the concept, and from these simulations predict that it is possible to engineer hardware with a 1024x1024 pixel array capable of performance [10 terapixels / second / Watt]. We have also performed proof-of-concept experiments with a toy-sized system (4x4 pixels), which achieves a performance [10 pixels / second / Watt], but validates that the randomness from C transmission is sufficient, and we have a reasonable expectation that if through foundry-scale efforts a state-of-the-art chip was produced, performance much closer to our theoretical/simulation predictions would be achieved.”

Based on the Reviewer suggestion, we have now revised the Discussion section and have added the following paragraphs in the revised manuscript:

In summary, we proposed a CMOS-compatible silicon photonics-based approach for large scale image processing. Our approach performs image compression and de-noising using an auto-encoder framework in which the first layer of the network is implemented using analog photonics, while the back-end image reconstruction is implemented using digital electronics. Our approach enables image compression with comparable quality to standard digital compression techniques such as JPEG and image denoising quality comparable to state-of-the-art digital neural network based de-noising algorithms. In contrast to the prevailing image processing approach, which performs a large number of image conditioning tasks at the front end, our approach is designed to compress the raw image data and use the neural network back-end to both reconstruct the original image and perform the image denoising and conditioning operations.

In this work, we presented a combination of numerical simulations and experimental characterization of a proof-of-principle device to validate this approach. Using numerical simulations, we optimized the system design by evaluating the compression quality as a function of kernel size and transmission matrix type (positive, real-valued vs. complex-valued). These simulations confirmed that this scheme can provide comparable quality compression to the JPEG algorithm. We also evaluated the impact of noise on the image compression quality and demonstrated the ability of this scheme to perform de-noising. Finally, we analyzed the potential for this scheme to perform high-throughput processing with low power consumption. For example, by processing 8x8 blocks of pixels (i.e. 64 inputs) in parallel at ~16 GHz, this approach could process 1 Terapixel/s. This would enable large format image sensors such as Gigapixel cameras to operate at speeds up to 1 kHz or standard Megapixel imaging systems to operate at frame rates up to 1 MHz. From a power-consumption perspective, we found that an optimized photonic encoder would consume 100X less energy per MAC than a GPU. In addition to the numerical simulations and analysis, we experimentally characterized a passive silicon photonic prototype with 16 inputs designed to process 4x4 pixel kernels and demonstrated compression using a real-valued experimental transmission (T) matrix. This confirmed that the properties of the experimental transmission matrix enable image compression with similar quality to the digital JPEG algorithm and image denoising with similar quality to digital electronic neural network based denoising algorithms. This experimental characterization also confirmed that this approach is robust to fabrication imperfections, provided calibration is performed after fabrication. Future work will focus on integration of high-speed modulators and detectors and increasing the kernel size to support 8x8 pixel blocks.